# Modelling Sustainable Urban Transport Performance in the Jakarta city Region: A GIS Approach

**Puji Adiatna Nadi ***  **and AbdulKader Murad ***

Department of Urban and Regional Planning, Faculty of Environmental Design, King Abdulaziz University, Jeddah 21589, Saudi Arabia

*   Correspondence: padiatna@stu.kau.edu.sa (P.A.N.); amurad@kau.edu.sa (A.M.);
    Tel.: +966-54-678-3750 (P.A.N.); Tel.: +966-54-100-9024 (A.M.)

**Abstract:** Measuring performance of Sustainable Urban Transport is an important effort to face the challenges of future trends. This study used Geographic Information System (GIS) application for modelling the performance of Sustainable Urban Transport (SUT) in the Jakarta city Region (JCR). The GIS applications include identifying the existing performance of SUT in Jakarta city, exploring the relationships between indicators of SUT, and producing a prediction model of SUT. Research methods used in this study were GIS techniques with geo-reference, classifications, polygon to raster, re-classifications, sum-weighted, ordinary least square (OLS), exploratory regression, and geographic weighted regression (GWR). The results revealed that the SUT model have more capability in measuring the performance of SUTs spatially and simply. This model is to visualize the effect of the indicator on the SUT performance and its influence, respectively. The results of this study also discovered that the JCR's Sustainable Urban Transport Performance was in medium level. The outputs of this modelling were useful for evaluating the level of SUT performances in the city based on districts area. Overall, this study provides valuable information on the SUT performance of the JCR, also highlights some important challenges faced in the future of SUT program development.

**Keywords:** sustainable urban transport; performance; indicator; modelling; GIS; Jakarta city

---

## 1. Introduction

Rapid urban growth has resulted in constant haphazard urban condition as well as land use and transportation issues, such as urban sprawl and congestion. The high economic activity of the society greatly affects the availability of the reliable urban transport. Problems in urban transportation are currently a concern, which urges various community groups to immediately find a solution, because of the serious impacts in the present and future. The rapid growth of private vehicles has led to many negative effects, such as congestion, air pollution, noise, accidents, and increased land consumption for transportation infrastructure. The quality of urban life needs to be prioritized for improvement in sustainability. Urban transportation is the main factor that requires performance measurement to determine the level of quality in providing services to users. Furthermore, specifically how the urban transport follows the concept of sustainable development needs to be found [1]. There are several ways to make performance measurements, one of them is by using certain indicators. The identification of the relationship between and among the critical sustainability indicators in a systematic approach will be meaningful research to pursue. The assessment is needed to go to the next stage in the form of management. Measurements of these indicators should be clearly quantifiable based on field data and outlined in spatial output. Identifying the relationship among indicators will help the researchers to understand clearly the chain of cause and effect of each sustainability issue [2]. Some spatial analytical issues in GIS on transportation have been explored by Miller [3]

such as modifiable areal units, boundary problems and spatial sampling, spatial dependency and spatial heterogeneity, and alternative representations of geographic environments. It provides a great knowledge for disseminating spatial analysis tools to transportation researchers.

It is very important to identify the methods and indicators as material to be analyzed for measuring the sustainable urban transport performance. Based on previous research of Nadi and Murad [4], there are a lot of methods in urban transport and sustainability concerns, such as framework analysis [5,6], decision support tools [7], Balanced Score Card (BSC) [8], backcasting scenario [9], policy evaluation [10], expenses evaluation [11], correlation analyses [12–14], dynamic optimization [15], and vector calculus [16]. Seeing the facts, the assessment of performance of the SUT depends on so many indicators, so that could be making it as weakness, i.e., the lack of priority in determining indicators. In addition, each selected indicator needs to clearly refer to each sustainability pillar and depict a typical scenario in the transportation system [17,18]. Hence, this study was conducted to find out the existence of the main indicators that were the parts of previous studies. Readiness of indicators is a foremost obstacle in the use of sustainability indicators because of national and regional characteristics [19]. For the scope of regional and local objectives, sustainable transportation consists of five indicators that have major impacts on three sustainability pillar, namely congestions (economy and social pillar), accidents (social and economy pillar), air pollution (environment and social pillar), noise pollution (environment and social pillar), and land consumption (economy and environment pillar) [5,20,21]. Therefore, the exploration focus of this study is in the five indicators to measure sustainable urban transport performance.

First, traffic congestion has a correlation with severity, duration, and spatial extent [22], meaning travel time or delay in excess of that normally under light or free-flow travel conditions. The objective of sustainability in this issue is to improve mobility and reliability on roads with Key Performance Indicators (KPIs); namely: travel time index, low traffic flow rate, high density of vehicles, and per capita congestion delay. Spatial Performance Indicators (SPIs) are the score of congestions based on area [23]. Second, the linkage of the traffic accident indicator to sustainability is about reducing the average number of accidents and the risk of accidents. The KPIs of traffic accidents report annual severe crashes per mile and the SPIs [24] described in the total of accidents divided by area. Third, the purpose of the air pollution indicator in the concept of sustainability is to reduce air pollution related to transportation activities. KPIs of this air pollution issue are daily PM10, $SO_2$, CO, and $NO_2$ emissions per mile of roadway. Meanwhile, its SPIs are manifested in traffic flows model, emission mapping, and concentration mapping [25]. Forth, the objective of noise pollution indicator on sustainability is to reduce traffic noise. KPI of traffic noise pollution is the buffer index in traffic noise levels. Meanwhile, SPIs of traffic noise pollution are calculated through the traffic noise gradient of the road segment [26]. Fifth, land consumption for transport infrastructure is conducted in relation to sustainability issues by optimizing mixed land-use to minimize transportation infrastructure and reduce land use in an effort to decrease the negative impact on the environment. KPIs for land consumption are Land-Use Balance and SPIs are the length or space of road network per district area [27]. Examples include acreage of sensitive lands on which new transportation infrastructure is built, and number of lane miles of roadways and a number of parking spaces are in park-and-ride lots [28].

There are several studies exploring the congestion indicators [19,29–35] have some research objectives such as reducing traffic congestion, reducing individual mobility, and reducing travel demand, and increasing the road capacity. Also, studies about traffic accident indicator [5,10,11,14,33,35–37] have research objectives such as reducing road accidents, providing traffic signs and markings using the proper standards and guidelines, and encouraging the traffic regulation and penalizing the violators. Studies about air pollution indicators [6,8,32,38] propose to reduce air pollution and quantifying air pollutant from energy consumption. The studies concern noise pollution indicators [8,39,40] and have goals to reduce of mobility of individual, to improve vehicle technology, and to reduce transport noise both at the source and through mitigation measures. Researches about land consumption for transport infrastructure [8,10,29,30,36,41] have several aims,

such as ensuring efficient use of land for transport infrastructure, to minimize the use of land for transport infrastructure, and to use Intelligent System Management to maximize the road infrastructure functions. Referring to the previous findings in this study, the research about the SUT is an important study to be examined more deeply. In addition, the study needs to be more focused in a quantitative research since the previous researches are mostly in qualitative. Therefore, as one of the main parts of the quantitative method, spatial analysis is important to analyze the indicators of the SUT. The challenge is about how to transfer the key performance indicators (KPIs) of the SUT into spatial analysis as Spatial Performance Indicators (SPIs). SPIs are then analyzed through the Geographic Information Systems (GIS) approach.

GIS applications have the ability to analyze various planning problems such as collecting, managing, analyzing, modelling, and presenting geographic or spatial data [42]. This capability makes it easier for planners to make decisions more effectively and efficiently. The growing problem of transportation requires effective treatments. The role of GIS as an analytical tool provides a major influence in providing solutions to solve transportation problems. GIS is a useful tool for transportation planners and urban planners to measure how much people or goods move as well as to measure, predict, evaluate, and monitor the extent of the transportation system in completing expected public goals [28]. The GIS benefits in data integration and map display by providing comprehensive evaluation and information with very precise and convincing output. Analysis of transportation systems with GIS is achieved through statistical analysis, diagrams, decision support systems, modelling, and databases, as well as coding, management, analysis, and reporting [43].

Previous researchers used various tools from GIS analysis to measure the level of performance in urban transportation by exploring information and analyzing it with certain scenarios that produced reliable research. For instance, geocoding with functions to create points on a map from a table of addresses in traffic air pollution analysis [44] and to produce a measurable acoustic parameter noise map [45]. In network analysis, this GIS tool is used to test transport policies in terms of emission effects at the link level and merged to the regional level [44], to measure the level of traffic accidents to present transport performances [43], and to measure safety level caused by traffic accidents [24]. In quantities analysis, the GIS tool functions to measure routes number per segment of road in measuring congestion [46,47]. In overlay analysis, the GIS function is to determine the congestion point according to the direction of the road [23]. In kernel density estimation (KDE), the analysis tool is to present the center of air pollution in area [44], to know the spread of accident risk [48,49], to identify dangerous locations on the road [50,51], and to calculate the probability density function of each crash site [52]. In K-means clustering, this GIS tool functions to create a road accident hotspot classification [48–50]. In spatial analysis, it has function to create sustainability metrics for the selection of transport infrastructure projects [53]. Spatial classification is used in modelling as a graph with a set of vertices and an arc [54].

One essential element for planners is having the ability to choose and adapt the model including in the transportation system [55]. Furthermore, models that simulate transportation systems and activity locations are usually referred to as land-use transportation interaction models that differ according to the size of the study area (urban, regional, and national) and the types of activities [56]. Hence, this study explores how to select indicators, transfer indicators from quantitative analysis into spatial analysis and conduct modelling. In the previous studies on transportation, many researchers have described several methods and tools in the process of research analysis using GIS [57]. One of most-used methods by previous researchers is the modelling method [58]. For instance, Son [59] conducted a study of calculating unit values from land based on the number and distance of the public transport station using spatial statistical methods in the form of correlation and regression analysis to make a calculation model that can predict the value of land units based on the number and distance with public transportation stations. Meanwhile, the modelling of the relationship between land use, transportation and energy using multiple regression and geometrically weighted regression (GWR) to develop land use and accessibility concepts has been explored by Kim [60]. Lopes et al. [61]

analyzed trips generation estimation by using spatial regression models in which the results were compared with multiple regression models for transportation demand forecast. Another study by Machado et al. [52] discussed traffic accidents by using spatial analysis to determine the location of accidents and their characteristics with a multivariate regression analysis (ordinary least squares/OLS) models and spatial regression models to investigate spatial correlations and spatial patterns and to understand the dynamics of traffic accidents. Whereas, this study will use GIS technique with geo-reference, classifications techniques, polygon to raster, re-classifications, sum-weighted, ordinary least square (OLS), exploratory regression, and geographic weighted regression (GWR).

The application of sustainable urban transport concept has begun to be initiated in Indonesia by encouraging the use of public transportation—Bus Rapid Transit (BRT)—in several provinces in Indonesia. The most advanced application of the concept of public transportation in Indonesia is Jakarta city as the capital of the state. There are some types of public transportation: public buses, BRT, trains, and MRT. The study of sustainable transport for Indonesia is still rare. Some of them are the result of cooperation between the Indonesian government and Germany in the form of a study of sustainable urban transport program in Indonesia, which includes a pilot project on the implementation of BRT in several regions in Indonesia, such as Medan, Batam, Palembang, Bogor, Jogjakarta, Solo, and Manado [62,63] and the relationship between the SUT development in Indonesia and climate change issues [64]. In recent times, the study of the assessment of the performance of SUT in Indonesia, especially Jakarta, has not been done.

Based on these findings, the following research questions arise. How are the current performances of sustainable urban transport in Jakarta city? How can GIS be used for model Sustainable Urban Transport performances in Jakarta city? Therefore, the purpose of this study is to model the basic indicators used to measure the performance of sustainable urban transport using the Geographic Information Systems (GIS) in Jakarta city. There are some sub-objectives of research: the first is related to identifying performance of each basic indicators of SUT in Jakarta city. The second is about measuring the level of SUT performance in Jakarta city. The third is about modelling the relationship between indicators in the five indicators of SUT. The structure of this study begins with an introduction which discusses in general the reason the authors conducted this study. Then, the methodology will describe about the study area, data collection, geodatabase model and analysis methods. Later, the results of the study are followed with a discussion and the last part is the conclusion.

## 2. Materials and Methods

### 2.1. Study Area

The research area encompasses Jakarta as a capital city of Indonesia and a city with high population in the world (see Figure 1). In 2017, the total population reached 10.37 million people with a population density of 15,663 people per km$^2$. The composition of Jakarta population in 2017 is that 50.15% are male and 49.85% are female. The average annual growth rate of the population of the city is 1.10% [65]. The Jakarta city consist of five municipalities consisting of 42 districts in land area and one regency as an island area. Based on the report from the Ministry of Transportation of the Republic of Indonesia [66], the total travel needs in DKI Jakarta are 21.9 million trips/day, while traveling by vehicle is about 15.3 million trips/day. The number of vehicles in 2016 was dominated by motorcycles—about 13.31 million units (73.92%)—while for passenger vehicles were 3.53 million units (19.58%), freights vehicles were about 0.69 million units (3.83%), bus cars were about 0.34 million units (1.88%), and other vehicles were about 0.14 million units (0.79%) with annual vehicle growth reaching 5.35% [67].

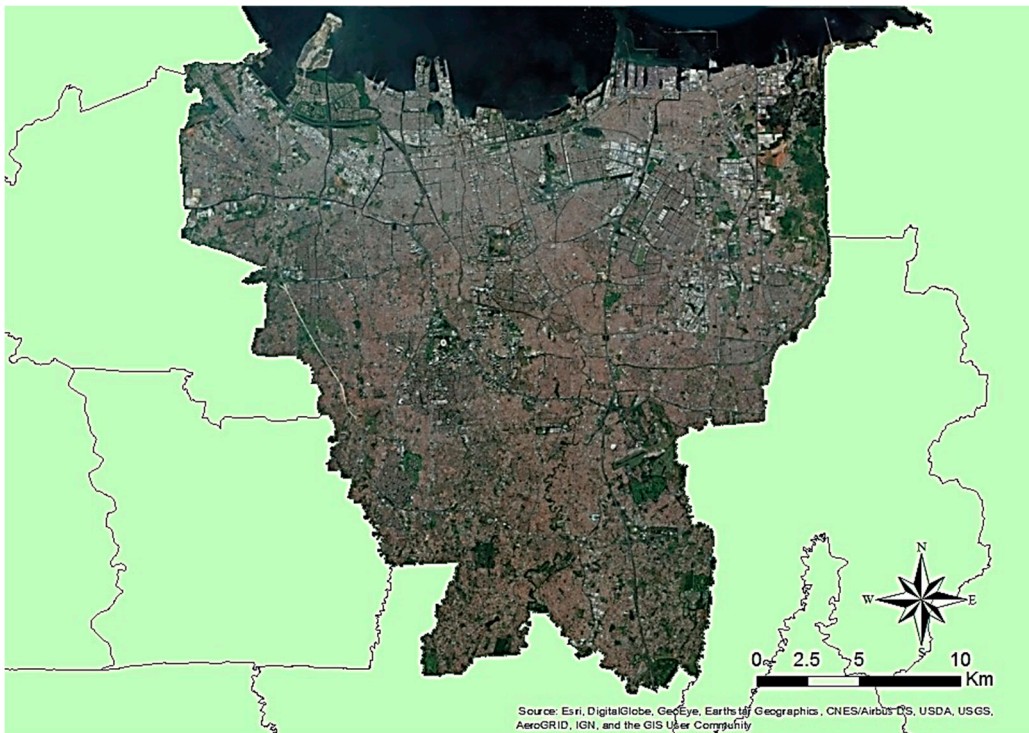

**Figure 1.** The Area of Study.

The Jakarta city region is surrounded by residential areas of the neighboring cities of Bogor, Depok, Tangerang, and Bekasi (Bodetabek) which are currently still growing. According to the Study on Integrated Transportation Master Plan for Jabodetabek (SITRAMP) [68] and Jabodetabek Metropolitan Priority Area (MPA) Strategic Plan 2012 [69], the transportation mode in Jakarta and Bodetabek area highly depends on road traffic (98%). Jakarta city has 8.1% of the road infrastructure in the Jabodetabek area with 13,720 km in length. The commuter trip demand to Jakarta city from Bodetabek area has significantly grown by about ten times in 18 years (1985–2002), and during the last eight years (2002–2010), it has been increasing by about 1.5 times. The central area of Jabodetabek has a lot of road links with an average speed of less than 20 km/h and the several sections with speed less than 10 km/h leading to serious congestion problems.

*2.2. Data Collection*

The data collections are retrieved from the Jakarta city Government, Indonesian Government, and other resources. Most map data were collected in .xls (Excel) format. For analysis in Arc GIS, these data were converted into ESRI shapefile. Several data are found in shape format in open data website and other resources. The following data were obtained from the official government and several sources are open data. Digital maps of Jakarta Boundary are in the form of GIS files, road centerlines, road network polygons, road names, and road types, collected from website address: gis.bpbd.jakarta.go.id. Demographic data, such as population number, population growth, total car number, total car ownership, total school, income per district, level of education, employed and unemployed, and transportation statistics are collected from the website: jakarta.bps.go.id. Transportation data, such as traffic congestion and traffic accident, are collected from police traffic corps office. Traffic air pollution and traffic noise pollution data are collected from Regional Environmental Agency office (see Table 1). Documents such as transportation master plan are from transportation ministry website and other sources.

**Table 1.** Data Collection of research.

| No. | Data | Type | Year | Source |
|---|---|---|---|---|
| 1 | District Boundaries | .shp | 2017 | GIS BPBD DKI Jakarta [70] |
| 2 | Demography | .xls | 2017 | Statistics Indonesia DKI Jakarta Province [71] |
| 3 | Road Network | .shp | 2016 | GIS BPBD DKI Jakarta [70] |
| 4 | Traffic Congestion | .xls | 2018 | Police Traffic Corps |
| 5 | Traffic Accident | .xls | 2017 | Police Traffic Corps |
| 6 | Traffic Air Pollution | .xls | 2017 | Regional Environmental Agency |
| 7 | Traffic Noise Pollution | .xls | 2018 | Regional Environmental Agency |
| 9 | Transportation Statistics | .pdf | 2016 | Statistics Indonesia DKI Jakarta Province [71] |
| 10 | Transportation Master Plan | .pdf | 2015 | Greater Jakarta Transportation Masterplan [72] |

The primary data are collected from the Regional Environmental Agency and Police Traffic Corps. The Police Traffic Corps data are mainly in the form of recording traffic accidents and traffic congestions that occur in Jakarta within a certain period of time. Records are equipped with information about the location of the incident, the type of event, the cause of the event and the time of the event. The Regional Environmental Agency data are in the form of field surveys at several measurement stations related to air pollution and noise pollution. This survey is carried out every day at several strategic locations in the city of Jakarta. Survey reports are packaged in the form of daily reports, monthly reports, and annual reports. Secondary data sources are obtained from the Statistics Bureau in the form of an overview of the Jakarta area about demography, socio-economic, transportation and others. In addition, secondary data is also obtained from online sources that are open to the public (open data) such as those obtained from the website of the Jakarta city government [73].

*2.3. Geodatabase Model*

The GIS arranges excellent instruments to obtain, collect, control, and exploration data, and unites great quantities of information from multi-disciplinary data sources. This study uses several methods of GIS approach to explore the performance of Sustainable Urban Transport in Jakarta city with five indicators and provided by ArcCGIS version 10.5.1 (ESRI, Redlands, US). This study has selected five basic indicators of sustainable urban transport data, namely traffic congestion, traffic accident, traffic air pollution, traffic noise pollution, and land consumption for transport infrastructure (see Figure 2). Identifying the relationship between indicators is very important for transport planners and urban planners. GIS helps describe and classify any type of features, such as points, lines and polygons based on values of attributes' data.

*2.4. The Classification Analyisis*

This study focuses on the measure of performance of Sustainable Urban Transport based on five basic indicators such as traffic congestion, traffic accident, traffic air pollution, traffic noise pollution, and land consumption for transport infrastructure. In transforming the quantitative form of five basic indicators as key performance indicators of SUT to spatial performance indicators (SPIs), several formulas are adjusted according to the types of indicators as provided in Table 2.

After the transformation phase from quantitative to spatial based on district units, the classification stage is proceeded. A variety weighting approach that evaluates the SUT performances for each district within the study area was used based on the previous research by Ramani et al. [74]. Each indicator is classified into five weights: High = 1 (lowest problems), Medium–High = 2, Medium = 3, Medium–Low = 4, and Low (highest problems) = 5 using natural breaks type of GIS classification tool (see Figure 3).

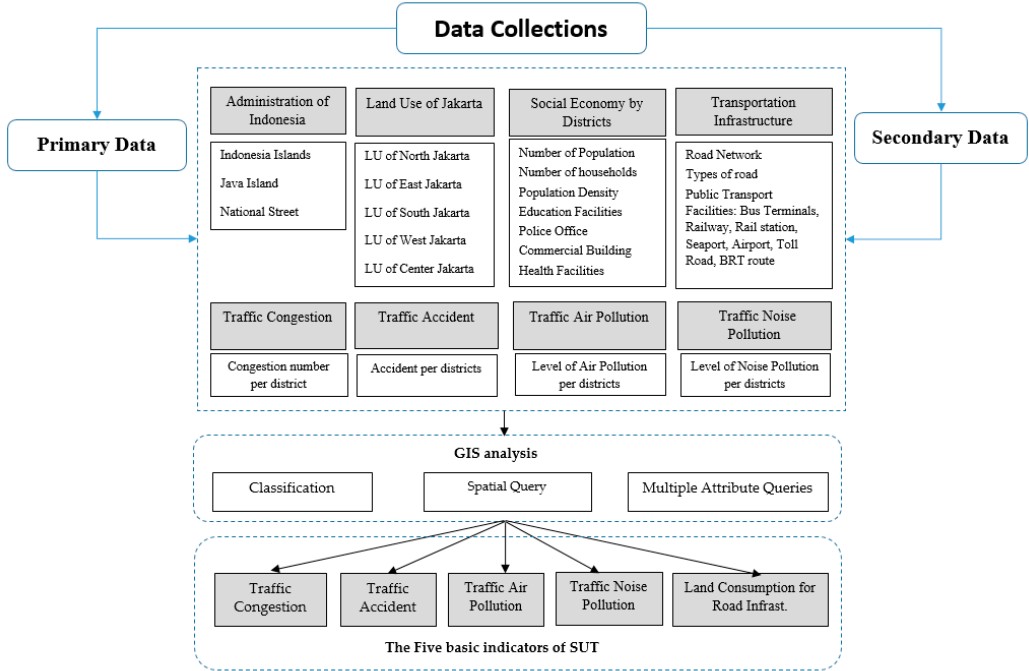

**Figure 2.** Geodatabase Model. SUT: Sustainable Urban Transport; LU: Land Use; GIS: Geographic Information System.

**Table 2.** Five Indicators of Index formulas.

| Five Basic Indicators of SUT | Equations |
| --- | --- |
| Traffic Congestions Indicator (TCI) | TCI = Total Congestion point/area (per district) |
| Traffic Accident Indicator (TAccI) | TaccI = Total Accident Number/area (per district) |
| Traffic Air Pollution Indicator (TAPI) | TAPI = The level of AQI [1]/area (per district) |
| Traffic Noise Pollution Indicator (TNPI) | TNPI = The level of NPI [2]/area (per district) |
| Transport Infrastructure in Land Consumption Indicator (TILCI) | TILCI = Total road area/District area |

[1] Air Quality Index; [2] Noise Pollution Index.

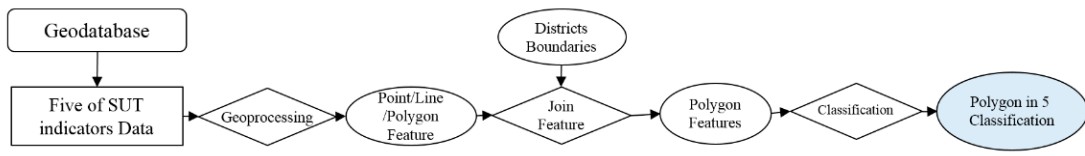

**Figure 3.** Classification Model.

## 2.5. Composite Index Analyisis

Some of the composite index literature at previous researchers have been carried out by Yigitcanlar et al. [75] developing Land Use and Public Transport Accessibility Index (LUPTAI) that measures and defines the accessibility of a location by developing a composite index of measures. It uses a series of indicators of accessibility for the purposes of quantification and GIS for the later analysis. Meanwhile, Gudmundsson and Regmi [76] state that it is necessary to decide how to weight each element in constructing a composite index. Important elements should have a higher weight, it may be determined by statistical analysis or correlations, or by the experts or by political or subjective choice. For this study, the final of SUTPI analysis uses default weighted with a little adjustment based on Ramani et al. [74] namely traffic congestion by 25%, traffic accident by 30%, traffic air pollution by

25%, traffic noise pollution by 10% and land consumption by 10%. Figure 4 shows the flowchart of Composite Index analysis using ArcGIS model builder.

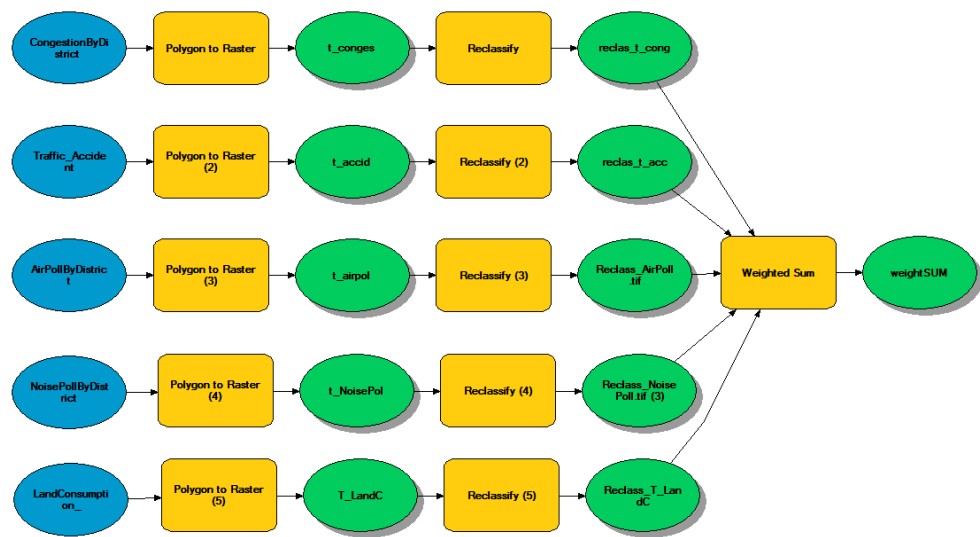

**Figure 4.** Geographic Information System (GIS) based Composite Index Model.

## 2.6. Spatial Statistical Analyisis

Spatial statistics is a set of exploratory techniques for describing and modelling spatial distribution, patterns, process and relationships consisting of coincidence, connectivity, area, proximity, orientation, length, and direction [77]. In GIS, spatial statistics is used to help assess patterns, trends, and relationships for better understanding behavior of geographic phenomena, pinpoint causes of specific geographic patterns, make decisions with higher level of confidence and summarize the distribution in a single number [78]. Spatial statistics tools in ArcGIS have several functions. First, it analyzes patterns (average nearest neighbor, high/low clustering/Getis-Ord General G), Multi-Distance Spatial Cluster Analysis/Ripley K Function, Spatial Autocorrelation/Morans I). Second, it maps clusters (Cluster and cluster analysis/Anselin Local Morans I), categorizes Analysis, Hot Spot Analysis/Getis-Ord Gi, Optimizes Hot Spot analysis, Optimizes outlier analysis, and similarity search. Third, it measures Geographic Distributions (Central Feature, Directional Distribution/Standard Deviational Ellipse, Linear Directional Mean, Mean Center, and Standard Distance). Forth, it calculates Distance Band from Neighbor Count, collects events, converts spatial weights matrix to matrix, and exports feature attribute to ASCII). Fifth, it models Spatial Relationship consists of Exploratory Regression, Generate Network Spatial Weights, Generate Spatial Weight Matrix, Geographically Weighted Regression/GWR, and Ordinary Least Squares/OLS (ArcGIS 10.5.1).

The important GIS tool in analysis based on spatial statistics is spatial autocorrelation analysis. This GIS tool is used to measure the correlation among neighboring interpretations in a pattern and the ranks of spatial clustering among neighboring districts. Moran's Index, in particular, has been used to study urban structure, complex urban growth and the intra-urban spatial distribution of socio-economic factors. Aljoufie [79] and Ji et al. [80] state that the Moran's Index test statistics is given by

$$I_M = \left( \frac{n}{\sum_i \sum_j W_{ij}} \right) \frac{\sum_i \sum_j W_{ij} \left( Y_{(R)i} - \overline{Y}_R \right) \left( Y_{(R)j} - \overline{Y}_R \right)}{\sum_j \left( Y_{(R)i} - \overline{Y}_R \right)^2} \tag{1}$$

where $W_{ij}$ is the component in the spatial weight matrix equivalent to district pairs i, j, and $Y_{(R)i}$ and $Y_{(R)j}$ are the different SUT variables (e.g., congestion, accident, air pollution, noise pollution and land consumption) for districts i and j with the mean variables expansion rate $\overline{Y}_R$. Because the weights are not row-standardized, the scaling factor $\frac{n}{\sum_i \sum_j W_{ij}}$ is applied. Moran's Index indicates the strength of the

spatial similarity or dissimilarity of neighboring districts. A positive Moran's I indicates the presence and degree of spatial autocorrelation.

Another important GIS tool in analysis based on spatial statistics is spatial regression analysis. This GIS tool is about modelling, examining, and exploring spatial relationships, as a statistical process for estimating the relationships among variables. In addition, it is useful to better understand the factors behind observed spatial patterns and to predict outcomes based on the understanding. The reasons to use regression analysis are to explore correlations, to predict unknown value, and to understand key factors [81]. The modelling of study uses three tools types, namely Ordinary Least Squares (OLS), Exploratory Regression, and Geographically Weighted Regression (GWR). OLS is a linear regression to generate predictions or to model a dependent variable in terms of its relationships to a set of explanatories. OLS as a global regression model has one equation for all features which is calibrated using data from all features, while the type of relationship is fixed. Exploratory regression tool is used to evaluate different arrangements of exploratory variables for OLS models that best clarify the dependent variables [82]. Whereas, GWR is a local form of linear regression used to model spatially varying relationships with each feature has one equation calibrated using data from nearby features; the type of relationships can vary across the study area [81]. Figure 5 describe OLS and GWR model analysis.

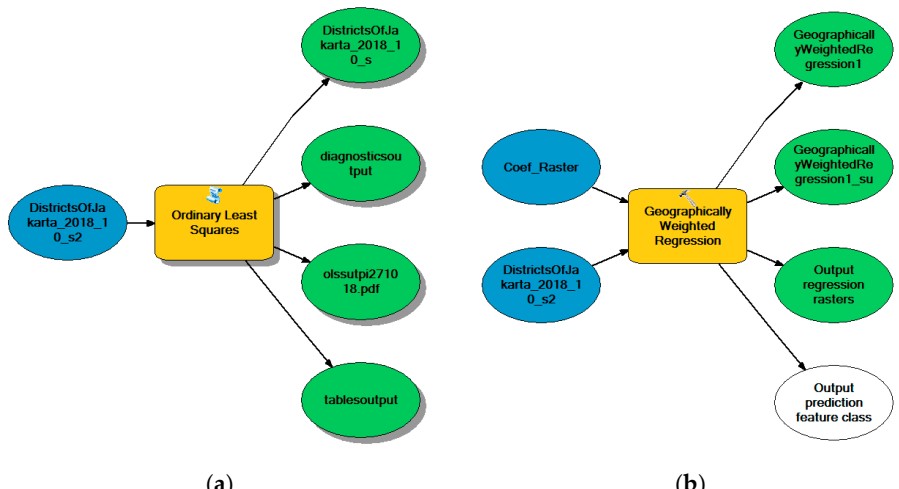

(**a**)                     (**b**)

**Figure 5.** (**a**) Ordinary Least Squares (OLS); (**b**) Geographically Weighted Regression (GWR). Source: ArcGIS 10.5.1.

Bennet and Vale [81] state that there are six steps of finding a properly specified model on spatial correlation report. First, each explanatory variable should have the relationship that the study expects (Check Coefficient (a)). Second, every explanatory variable has an asterisk (star sign) (check the probability (b) and robust_Pr (b)), which means the coefficient has an expected sign and is statistically significant. Third, test the clustering of the residuals using the Spatial Autocorrelation tool. It means that the residuals (e) should not be clustered in location or in value. Forth, verify that residuals are normally distributed using the Jargue-Bera test (make sure the Jarque-Bera statistic does not have an asterisk) meaning not bias. Fifth, each variable should tell a different part of the story (variance inflation factor/VIF values lower than 7.5). Its objective is to avoid redundant variable (among explanatory variables). Sixth, evaluate model performance (Adjusted R-Squared (d) more than 0.5). It means that the stronger adjusted R2 the better model performance.

*2.7. Regression Models for Sustainable Urban Transport Performance*

Ordinary Least Square (OLS) and Geographically Weighted Regression (GWR) are regression analyses that allow modelling, examining, and exploring spatial relationships. The function is to

explore spatial variation. It used to better understand the factors behind observed spatial patterns, and to predict outcomes based on that understanding.

Some terminology in regression models:

1. Dependent variable (Y): variable to model or predict
2. Explanatory variables (X): variable that influence or help explain the dependent variable
3. Coefficient (β): values, computed by the regression tool, reflecting the relationship and strength of each explanatory variable to dependent variable
4. Residuals (ε): the portion of the dependent variable that is not explained by the model; the model under and over predictions.

$$Y = \beta_0 + \beta_1 X_1 + \beta_2 X_2 + \beta_3 X_3 + \ldots \beta_n X_n + \varepsilon \tag{2}$$

This study models five basic indicators of Sustainable Urban Transport Performance using spatial statistics as shown in Figure 6.

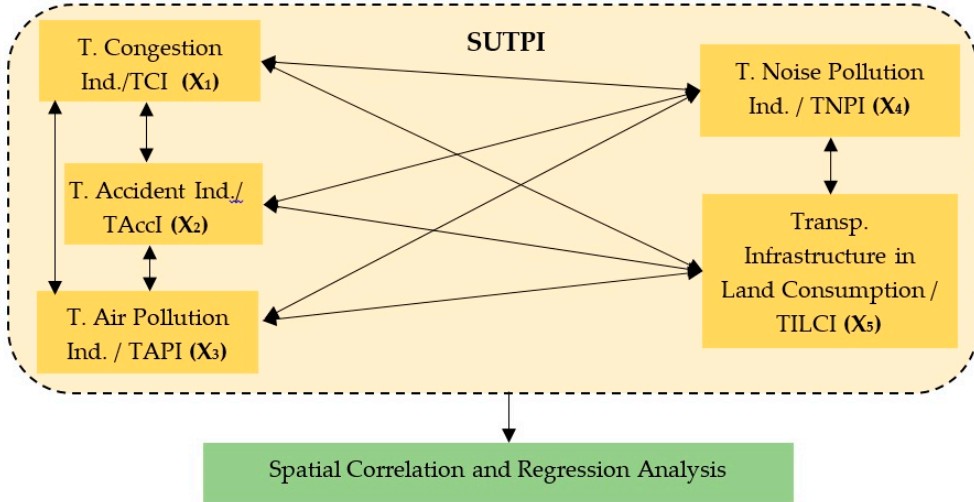

**Figure 6.** Modelling Relationship of Sustainable Urban Transport Performance Index (SUTPI) indicators.

Therefore, this study proposes the regression models for Sustainable Urban Transport Performance Index (SUTPI) as follows.

$$\text{SUTPI} = \beta_0 + \beta_1(\text{TCI}) + \beta_2(\text{TAccI}) + \beta_3(\text{TAPI}) + \beta_4(\text{TNPI}) + \beta_5(\text{TILCI}) + \varepsilon \tag{3}$$

where SUTPI as dependent variable (Y): variable to model or predict, five basic indicators as explanatory variables (X): variable that influence or help explain the dependent variable. All steps of GIS analysis are illustrated in Figure 7.

*2.8. Model Validation*

As a necessary step in modelling methodology, in order to validate the model, the simulation data and the historic data should be used for specific years [83]. Noviandi et al. [84] state that there are two ways to do model validation. First, it uses a comparison chart between simulation data and statistical data, done by integrating the representation of empirical data with the simulation results of data in one graph to represent the behavior of the occurred phenomena. Second, it runs a statistical test by measuring the results of simulation errors with the mean error method to reflect the level of precision and accuracy of the model. The data collected, after suitable processing, can be used as inputs to performance and user cost models. The data collected can also be used for monitoring purposes to validate model predictions and update a model system after it has been implemented [85]. For model validation in this study, SUTP model is validated by comparing model results of a historic period with

the actual changes of SUTP [86]. SUTP change model is validated by comparing predicted results of model with the actual result of SUTP. To validate SUT model using GIS in study, as adopted from Guan et al. [86], it compares the real data value with the predicted value, respectively. It is evident that the simulated values of variables are closer to the real value; it suggests the reasonability of model. Secondly, the simulated values of variables have the low relative errors of ±5% compared to the real values.

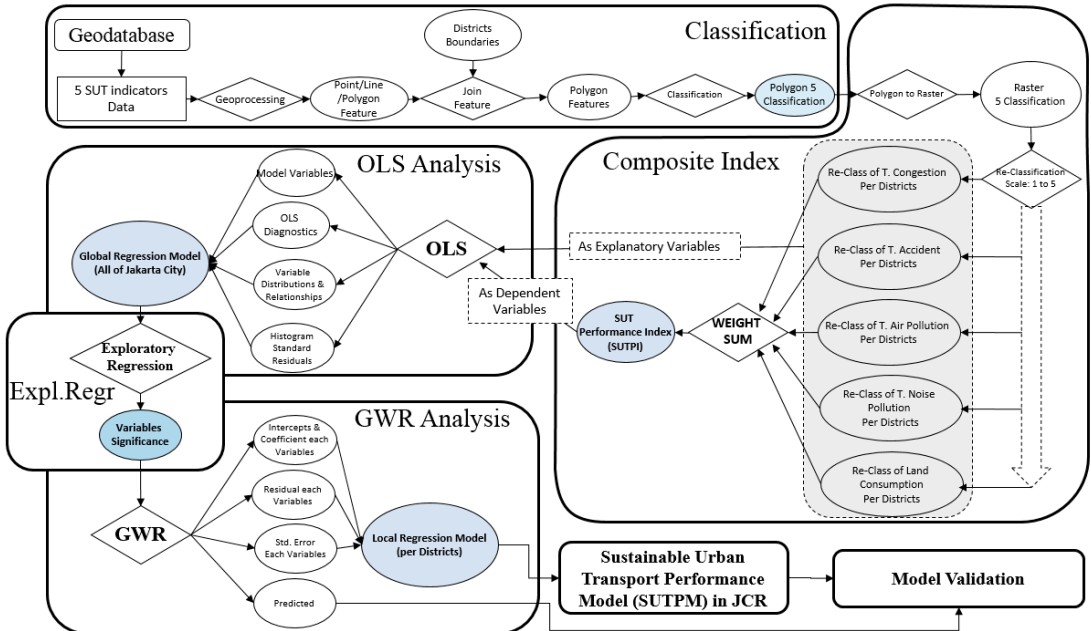

**Figure 7.** Modelling Relationship of SUTPI indicators.

### 2.9. GIS Analysis Flowchart

Figure 7 describes all GIS analysis processes in this study. First, classification analysis stage classifies the five indicators of SUT into five classifications: high, medium-high, medium, medium-low, and low based on districts. Then, composite index analysis combines the values of the five basic indicators of SUT through weighting and produces an index as the value of SUT performance per district. Next, OLS analysis is a global regression analysis of SUT performance by making five SUT indicators as independent/explanatory variables and the composite index results as a dependent variable that produces an equation model for the Jakarta region as a whole (global). Next, exploratory regression is an analysis that tests the independent variables influencing the model, significant or not, positive or negative. Next, GWR is a local regression analysis, after going through the testing process at exploratory regression, the variables that are significant to the model are identified. Then, the independent variables that pass the test are used as input so that a reliable model can be obtained. Finally, the validation model is a comparison between the initial value of the SUT model and the predictive value (the output value of the GWR analysis) whether it has similarities or even has distant differences. If it is similar, it is said that the model is robust, and vice versa.

## 3. Results and Discussion

### 3.1. Identifying Five Indicators of SUT

Indicators can be used as an effective tool in quantifying and analyzing the performance of sustainable urban transport [2]. Five Indicators were selected and developed in this study to quantify the situation of sustainable urban transport as analysis method in GIS. Figure 8 describes the performance of five basic indicators using five classifications from highest to lowest (green to red) in the Jakarta city region.

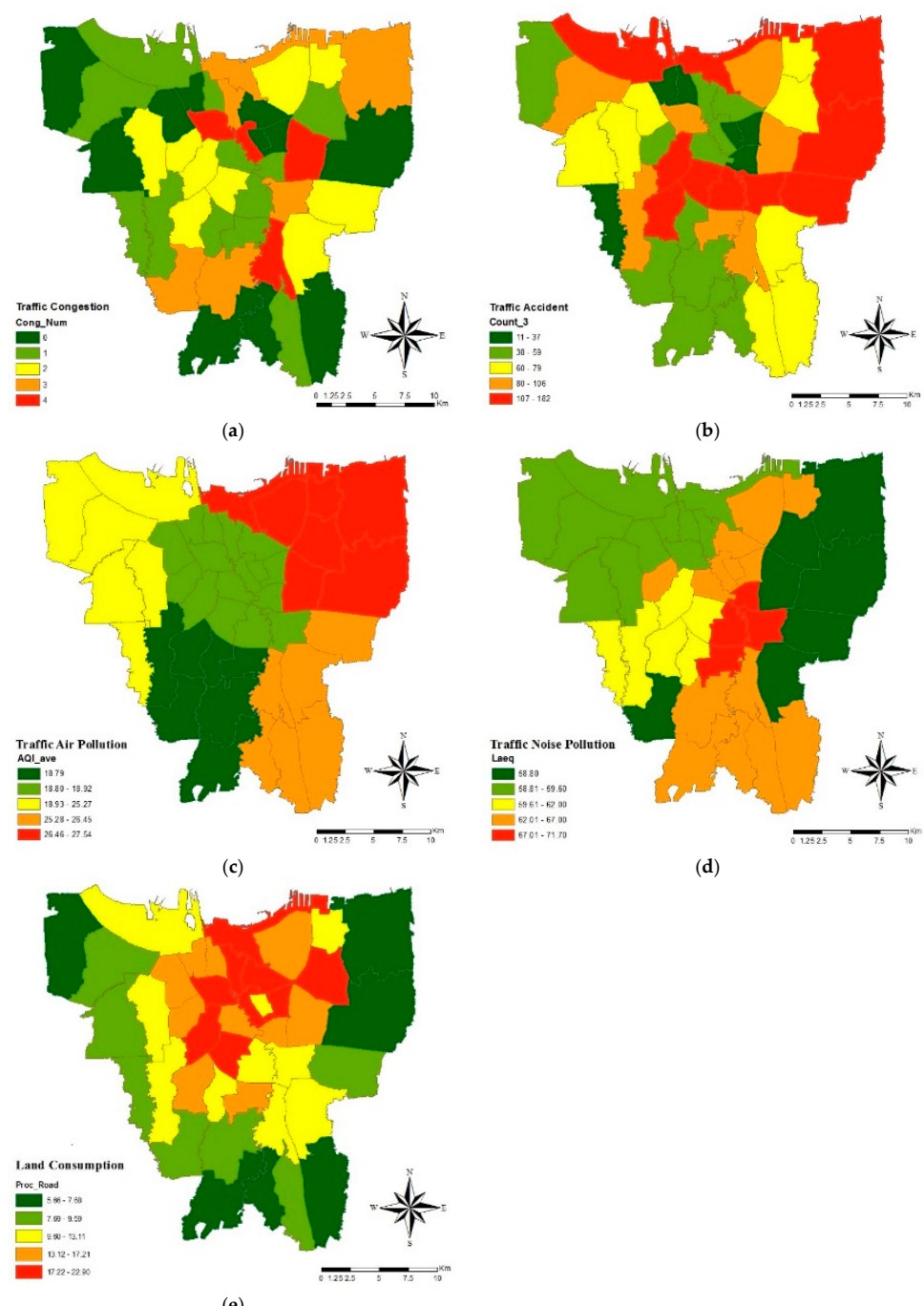

**Figure 8.** Classifications of Five Basic Indicators of SUT. (**a**) Traffic Congestion Indicator (TCI); (**b**) Traffic Accident Indicator (TAccI); (**c**) Traffic Air Pollution Indicator (TAPI); (**d**) Traffic Noise Pollution Indicator (TNPI); (**e**) Land Consumption Indicator (TILCI).

Traffic congestion density depicts high SUT performance in different districts throughout Jakarta city. Low traffic congestion density in a district means high performance. Figure 8a shows a classification model for traffic congestion number based on districts. It describes the performance of traffic congestion indicator in Jakarta city. It is noted that low performance districts concentrate at the central and north Jakarta. On the other hand, high performance districts spread all over city urban boundaries. It shows that there is an excessive concentration of vehicles in the middle of the city. The middle of the city as a center of office, business and government administration is a pulling factor for various community activities resulting in a concentration of movement in the area. Figure 8b

shows the spatial classification of the absolute scores of traffic accident indicator. The highest traffic accident in 2017 is in Tebet district with 182 incidents, Duren Sawit district with 173 incidents, and Setia Budi districts with 172 incidents. Traffic accidents often occur in the northern, central, and eastern parts of Jakarta. The North Jakarta area has high transportation activities with the existence of a national international sea port for the movement of large vehicles for loading and unloading of goods. The middle part of Jakarta is filled with business activities such as the Tanah Abang market. The east part of Jakarta is also a business center, commuters' flow, and industry center, so that the movement of the community is high in this area.

Figure 8c shows that high performance is dominant with this indicator as compared with medium and low performance reflected by the number of districts. It is noted that high performance districts concentrate at the south city. On the other hand, low- and middle-performance districts spread in East and West Jakarta. Air pollution caused by transportation activities has a link in the concept of sustainability with land use and traffic activity in the following variables: traffic, population and land use. Based on the data retrieved from the Regional Environment Control Agency (BPLHD) of DKI Jakarta, until 2011, there were no less than 6 million motorcycles passing every day on the streets of Jakarta. Based on recent data, there are six units of the noise control station in Jakarta city. The analysis using GIS through data entry which was collected to attribute and then in overlay with district file shape obtained results as follows. Traffic noise pollution level depicts high SUT performance in different districts throughout Jakarta city. It is found that low and medium performance districts concentrate at the city center and in some districts drawn to urban peripheries. On the other hand, high performance districts concentrate at the east of city as shown in green color. This detailed result can be seen using the classification method as illustrated in Figure 8d.

Land consumption for road infrastructure becomes the highest consumption of land for transportation infrastructure in Sawah Besar district with a land use rate of 22.9%, Gambir district by 22.11% and Tanah Abang district by 20.59%. Land consumption for transport infrastructure depicts medium SUT performance in different districts throughout Jakarta city. Low land consumption for transport infrastructure level in a district means the high performance. Figure 8e shows that medium until low performance is dominant in Jakarta based on districts. It is noted that high performance districts spread at the city urban peripheries. On the other hand, low performance districts concentrate at the city center. Table 3 depicts the five basic indicators in measuring the SUT performance in Jakarta city Region based on districts and areas as result from classification and composite index analysis as shown in Figure 4.

**Table 3.** The classification of five basic indicators to measure the SUT performance based on districts.

| Performance | TCI | | TaccI | | TAPI | | TNPI | | TILCI | |
|---|---|---|---|---|---|---|---|---|---|---|
| | Dist. | Area | Dist. | Area | Dist. | Area | Dist. | Area | Dist. | Area |
| High | 11 | 192.19 | 6 | 35.46 | 7 | 112.49 | 7 | 175.20 | 6 | 177.65 |
| Medium–High | 12 | 167.08 | 11 | 145.34 | 16 | 110.19 | 11 | 183.35 | 10 | 95.81 |
| Medium | 9 | 129.07 | 8 | 145.35 | 6 | 145.37 | 6 | 72.15 | 11 | 148.68 |
| Medium–Low | 6 | 114.94 | 7 | 108.93 | 6 | 114.27 | 14 | 178.99 | 10 | 95.81 |
| Low | 4 | 39.80 | 10 | 208.02 | 7 | 160.78 | 4 | 33.40 | 9 | 82.58 |

*3.2. Measuring the Performance of SUT*

The measurement of Sustainable Urban Transport Performance in Jakarta city Region is based on five basic indicators such as Traffic Congestion, Traffic Accident, Traffic Air Pollution, Traffic Noise Pollution, and Transport Infrastructure in Land Consumption. A variety weighting approach that evaluates the SUT performances for each district within the study area was used. Figure 9 shows the output of composite index analysis of SUT performance in Jakarta city. The results of the SUT performance composite index analysis, which combines the five basic indicators, show the patterns of sustainable city transportation performance that are spread and collected in the city of Jakarta. It was

found that SUT high-performance districts were scattered in the Central, West, and South Jakarta. Low performance is concentrated and clustered in the North and East of the city. It is significant that districts with intermediate level performances are dominant and spatially separated.

In particular, around 13.76% of the Jakarta area is at a high level of SUT performance, around 22.22% is at the middle-high level of performance, around 15.24% of the Jakarta area is at a moderate level of performance, around 30.40% of the area Jakarta is at the low-middle level, around 18.38% of the Jakarta area is at a low level of performance. SUT performance measurement is important to evaluate the transportation system so that it is more efficient to create a competitive urban area that is able to provide easy access for the people to work, education and health facilities. GIS has exposed the ability to group, visualize and calculate SUT performance. This index facilitates urban transport planners in identifying the level of SUT performance in Jakarta and makes it easier to determine policies in improving sustainability in a region [87]. Table 4 depicts the result of composite index analysis as shown in Figure 9.

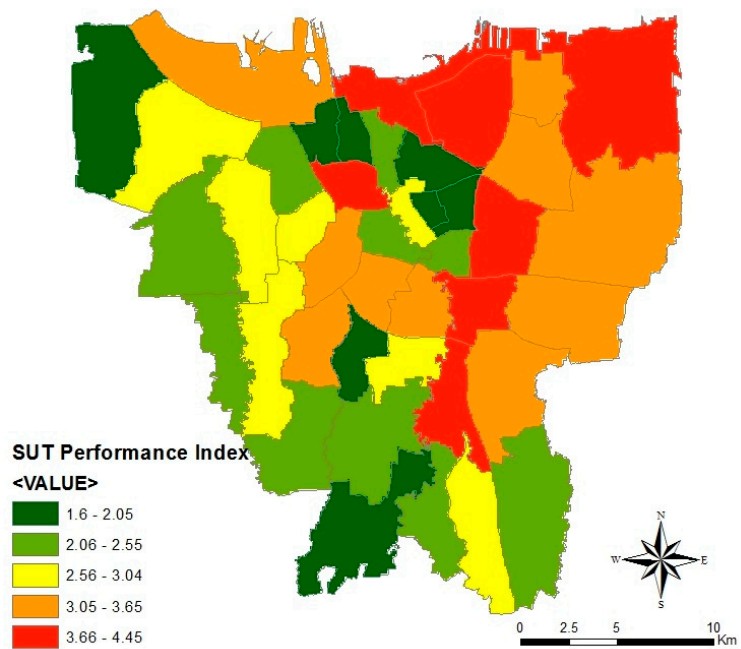

**Figure 9.** SUTPI in Jakarta based on GIS Composite Index Analysis result.

**Table 4.** SUTPI Performance in Jakarta city.

| Performance | SUTPI | | District Name |
|---|---|---|---|
| | Total Dist. | Area (sqkm) | |
| High | 8 | 90.28 | Cempaka Putih, Jagakarsa, Johar Baru, Kemayoran, Kalideres, Mampang Prapatan, Taman Sari, Tambora |
| Medium–High | 10 | 145.76 | Cilandak, Cipayung, Grogol Petamburan, Kembangan, Matraman, Menteng, Pasar Minggu, Pasar Rebo, Pesanggrahan, Sawah Besar |
| Medium | 7 | 99.99 | Cengkareng, Ciracas, Kebayoran Lama, Kebon Jeruk, Palmerah, Pancoran, Senen |
| Medium–Low | 10 | 199.43 | Cakung, Duren Sawit, Kebayoran Baru, Kelapa Gading, Koja, Makasar, Penjaringan, Setia Budi, Tanah Abang, Tebet |
| Low | 7 | 120.59 | Cilincing, Gambir, Jatinegara, Kramat Jati, Pademangan, Pulo Gadung, Tanjung Priok |

### 3.3. Development Model of SUT Performance in Jakarta City

Based on the composite index results, regression modelling is used to determine the extent of the relationship between indicators and SUT performance. Several steps of model regression analysis of sustainable urban transport are used in this study as follows:

### 3.3.1. Descriptive Statistical Analysis

As the dependent variable, Sustainable Urban Transport Performance Index (SUTPI) had mean value approximately 2.98 (almost 3, meaning in the middle performance level) and standard deviation of 1.42. For the explanatory variable i.e., the mean of traffic congestion indicator was about 2.67 point per district with standard deviation of 1.32. The mean value of traffic accident indicator was 3.14 and standard deviation of 1.39. For traffic air pollution indicator had mean value was 2.76 and standard deviation of 1.34. Traffic noise pollution indicator had mean value of 2.88 with standard deviation of 1.26. Land consumption indicator had mean value of 3.21 and standard deviation of 1.34 (see Table 5).

**Table 5.** Descriptive Statistics (based on classification scale from 1 to 5).

| Indicators | Mean | Standard Deviation | Number | Sum |
|---|---|---|---|---|
| T. Congestion | 2.666667 | 1.321375 | 42 | 112 |
| T. Accident | 3.142857 | 1.389954 | 42 | 132 |
| T. Air Pollution | 2.761905 | 1.34181 | 42 | 116 |
| T. Noise Pollution | 2.880952 | 1.257404 | 42 | 121 |
| Land Consumption | 3.214286 | 1.336942 | 42 | 135 |
| SUTPI | 2.952381 | 1.379309 | 42 | 124 |

Based on descriptive statistical analysis for traffic congestion indicators, it is identified that 52.03% of Jakarta city area are under the mean value (2.67), 20.71% area are in the mean value, and 27.26% are above the mean value. It describes that the number of districts in Jakarta city with the performance above the average of TCI value were dominant at 52.03%. In traffic accident indicators, it is identified that 27.47% of Jakarta city area are under the mean value (3.14), 22.20% area are in the mean value, and 50.33% are above the mean value. It means that the number of districts in Jakarta city with the performance level under the average of TAccI value were dominant with 50.33%. In traffic air pollution indicators, it is identified that 34.27% of Jakarta city area are under the mean value (2.76), 23.94% area are in the mean value, and 41.79% are above the mean value. It illustrates that about 41.79% of districts in Jakarta have the performance level under the average of TAPI value. In traffic noise pollution indicators, it is identified that 56.13% of Jakarta city area are under the mean value (2.88), 14.26% area are in the mean value, and 29.61% are above the mean value. It indicates that about 56.13% of districts in Jakarta city were above the average of TNPI value. In land consumption indicators, it is identified that 48.86% of Jakarta city area are under the mean value (3.21), 23.87% area are in the mean value, and 27.27% are above the mean value. It means that the number of districts in Jakarta city with a performance level under the average of TILCI value were dominant at 48.86%. In the case of the SUTP index, it is revealed that 35.98% of the Jakarta city area are under the mean value (2.95), 15.24% are in the mean value, and 48.78% are above the mean value. It means that about 48.78% of districts in Jakarta city have the performance level under the average of SUTPI value.

### 3.3.2. Scatter Plots Analysis

Figure 10 shows the relationship between indicator (as exploratory variables) and with SUTPI (as dependent variable). Scatter plot of SUTPI and TCI indicates a positive relationship as shown in Figure 10a. Its means there is a positive relationship between SUTPI and TCI. The slopes of this plot (+) indicates that as the number of TCI increases, SUTPI value decreases. In addition, since the *p*-value is less than 0.05, the relationship is significant. The slope of SUTPI versus TAccI as shown in Figure 10b indicates that there is a positive relationship between SUTPI value and number of Traffic

Accident. Its means that as the performance SUTPI increases, traffic accident increase. In addition, since *p*-value is less than 0.05, it also shows a significant relationship between SUTPI and traffic accident. The relationship between SUTPI and TAPI shows as a positive relationship between SUT Performance and Traffic Air Pollution. The coefficient represents that as Traffic Air Pollution Indicator increases, SUT Performance increases. Furthermore, since the *p*-value is less than 0.05, it shows a significant relationship between them (see Figure 10c).

The scatter plot for SUTPI and TNPI shows a negative relationship between SUT performance and Traffic Noise Pollution as shown in Figure 10d. Its means the coefficient represents that SUTPI increases as the traffic noise pollution decreases. Since p-value is less than 0.05, it proves that is a significant relationship between SUTPI and TNPI. Figure 10e also represents the relationship between SUTPI and TILCI which shows a positive relationship between SUT performance and Land consumption for road infrastructure. In other words, as the SUTPI increases, the performance of land consumption increases as shown in the coefficient. Since *p*-value is less than 0.05, it proves that there is a significant relationship between them.

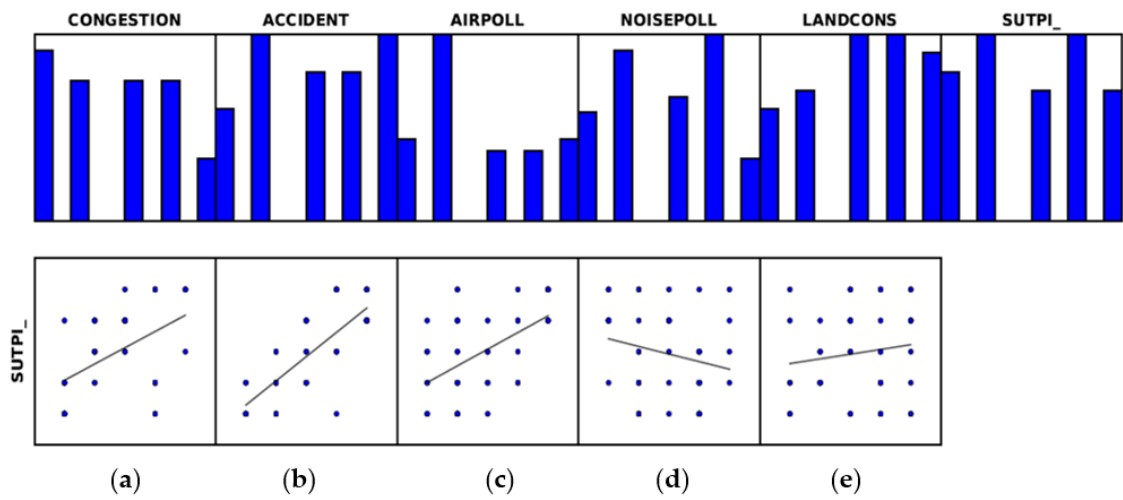

**Figure 10.** Variable distributions and relationships. (**a**) SUTPI and TCI; (**b**) SUTPI and TAccI; (**c**) SUTPI and TAPI; (**d**) SUTPI and TNPI; (**e**) SUTPI and TILCI.

### 3.3.3. The Ordinary Least Square (OLS) Analysis (Global Regression Model)

The result of multiple linear regression analysis with original dependent variable shows that a model adjusted R-square value of 0.873. In addition, *p*-value is less than 0.05 in the OLS test which proves that the model is significant. This table shows only variable with *p*-value was lower than 0.05 that is significant predictor for the SUTPI model. TILCI has *p*-value more than 0.05, it means it is not significant predictor for SUTPI model. Since the Variance Inflation Factor (VIF) ranges from 1.13 to 1.343 (under 7.5), there is no multi-collinearity problem in the predictive model. Table 6 shows the result of multiple linear regression using the ordinary least square (OLS) as global regression model in Jakarta city.

It is important to check the model residuals that are normally distributed or not, before the next interpretation step. In this study, spatial autocorrelation analysis report for standard residual indicates that the model residual is not clustered. It means the model has good performances (see Figure 11).

Based on the coefficient analyzed in this study, the formula is developed to analyze how the SUT performance can be affected by five basic indicators of SUT (see Equation (3)). In addition, about 85.5% variability of the SUTPI can be explained by the multiple linear regression.

$$\text{SUTPI} = -1.815 + 0.398\text{TCI} + 0.625\text{ TAccI} + 0.391\text{ TAPI} + 0.124\text{ TNPI} + 0.094\text{ TILCI} \qquad (4)$$

**Table 6.** The coefficients of SUTPI Model.

| Variable | B (Coefficient [a]) | Std. Error | t-Value (t-Statistic) | p-Value (Robust_Pr [b]) | VIF |
|---|---|---|---|---|---|
| Intercept | −1.815 | 0.456 | −3.982 | 0.000003 * | - |
| TCI | 0.398 | 0.067 | 5.971 | 0.000005 * | 1.155 |
| TAccI | 0.625 | 0.063 | 9.973 | 0.000000 * | 1.130 |
| TAPI | 0.391 | 0.071 | 5.514 | 0.000000 * | 1.343 |
| TNPI | 0.124 | 0.073 | 1.696 | 0.033603 * | 1.260 |
| TILCI | 0.094 | 0.066 | 1.432 | 0.194586 | 1.152 |

* Statistically significant.

The estimated coefficient analyzed in the regression analysis are as follows:

- $\beta_1$ is 0.398 which means that one index of Traffic Congestion causes a value increase of 0.398 in the SUTPI

- $\beta_2$ is 0.625 which means that one index of Traffic Accident causes a value increase of 0.625 in the SUTPI

- $\beta_3$ is 0.391 which means that one index of Traffic Air Pollution causes a value increase of 0.390 in the SUTPI

- $\beta_4$ is 0.124 which means that one index of Traffic Noise Pollution causes a value increase of 0.124 in the SUTPI

- $\beta_5$ is 0.094 which means that one index of Land Consumption causes a value increase of 0.094 in the SUTPI

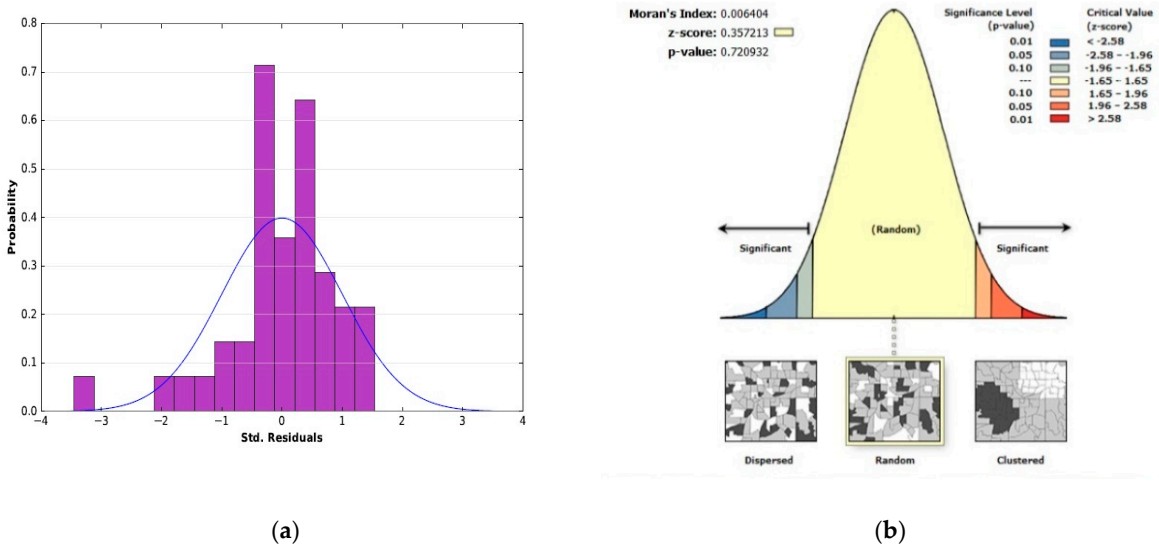

**Figure 11.** (**a**) Standard Residuals of OLS SUTPI (**b**) Non-clustered Residual.

Traffic accident indicator is the most significant cause value of the SUTPI, followed by traffic congestion indicator, traffic air pollution, traffic noise pollution, and land consumption, respectively. There was no multi-collinearity problem of the predictive model since the Variance Inflation Factor (VIF) ranged under 7.5. Furthermore, the adjusted R-square of the transformed model was 0.855 which indicated that 85.5% variability in SUTPI value could be explained by these variables.

3.3.4. The Exploratory Regression Analysis (Variables Test)

The exploratory regression results based on the highest adjusted R-Squared [82] describe that the SUTPI model had AdjR2 (Adjusted R-Squared) value of 0.855 as good performance, AICc (Akaike's

Information Criterion) value of 76.91, JB value of 0.00, K(BP) (Koenker (BP) Statistic *p*-value) value of 0.50, VIF (Variance Inflation Factor) value of 1.34 (under 7.50 is good), SA (Global Moran's I *p*-value) value of 0.92. The results of variable significance indicate that congestion, accident, and air pollution indicators had 100% significant. Whereas, the noise pollution indicators were not significant and land consumption indicators just had 18.75% significant (see Table 7).

**Table 7.** Summary of Variable Significance.

| Variable | % Significant | % Negative | % Positive |
|---|---|---|---|
| Traffic Congestion | 100.00 | 0.00 | 100.00 |
| Traffic Accident | 100.00 | 0.00 | 100.00 |
| Traffic Air pollution | 100.00 | 0.00 | 100.00 |
| Transport Infra. in Land Consumption | 18.75 | 0.00 | 100.00 |
| Traffic Noise Pollution | 0.00 | 62.50 | 37.50 |

3.3.5. The Geographically Weighted Regression GRW analysis (Local Regression Model)

Table 8 describes the result of GWR analysis for SUTPI as dependent variable and five basic indicators of SUT as explanatory variable. It also shows that R2 Adjusted value is 0.845 which means that the SUTPI model could be explained by these variables.

**Table 8.** GRW Analysis Result.

| ID | Variable Name | Variable Value |
|---|---|---|
| 0 | Neighbors | 42 |
| 1 | Residual Squares | 9.149773 |
| 2 | Effective Number | 11.677675 |
| 3 | Sigma | 0.549318 |
| 4 | AICc* | 84.768823 |
| 5 | R2 | 0.885492 |
| 6 | R2Adjusted | 0.845169 |

* AICc: Akaike's Information Criterion

The analysis pattern with spatial autocorrelation (Moran-I) indicates that residual was not significantly distributed (random). Figure 12 shows that residual of the GWR model for SUTPI is not clustered in location or in value. It means that the model showed good performance. The GWR results describe some findings related to SUT performance, such as districts with the highest LocalR2 values—the Ciracas district and Cipayung district (0.872707)—and the lowest LocalR2 value was Kali Deres district (0.872691). The highest coefficient value for the congestion indicator was Cilincing district (0.398303), and the lowest was in the Jagakarsa district (398255). The highest coefficient value for traffic accident indicator was Kali Deres district (0.625547), and the lowest was in Cilincing district (0.625296). The highest coefficient value for the traffic pollution indicator was Cilincing district (0.390725), and the lowest was in Jagakarsa district (0.390424). The highest coefficient value for pollution noise traffic indicators was the Deres Kali district and Penjaringan district (0.124349), and the lowest one was in the Ciracas district (0.124108). The highest coefficient value for the indicator land consumption was Ciracas district (0.094348), and the lowest one was in Penjaringan district (0.094179) (see Table 9).

This part describes the coefficient in each district based on the GWR analysis results and how to interpret them [81]. Figure 13a shows that Traffic Congestion Indicator (TCI) clustered was more significant toward alteration of SUTPI value in North-East of Jakarta. The classification used standard deviation (SD) method in order to identify the features (districts) were above or below the average coefficient value. The coefficient value indicates that the highest influence of the SUT from the congestion indicator in the North-East Jakarta district. This indicates that congestion prevention is very important to be focused in this area. As recommendation, the government's strategic programs in breaking down congestion need to be considered in the north-east area of Jakarta as the region

with the most significant impact on congestion. Thus, the policies adopted such as the use of public transport and traffic arrangements from and to the sea port need special attention. Figure 13b describe that traffic accident indicator (TAccI) was clustered more significantly toward alteration of SUTPI value in South-West of Jakarta. The west area of Jakarta had high coefficient value for traffic accident variable than others. As recommendation, the strategies to reduce traffic accident rates are prioritized in the West of Jakarta area because, based on the results of the regional regression analysis, the highest coefficient will increase the influence on SUT performance in Jakarta.

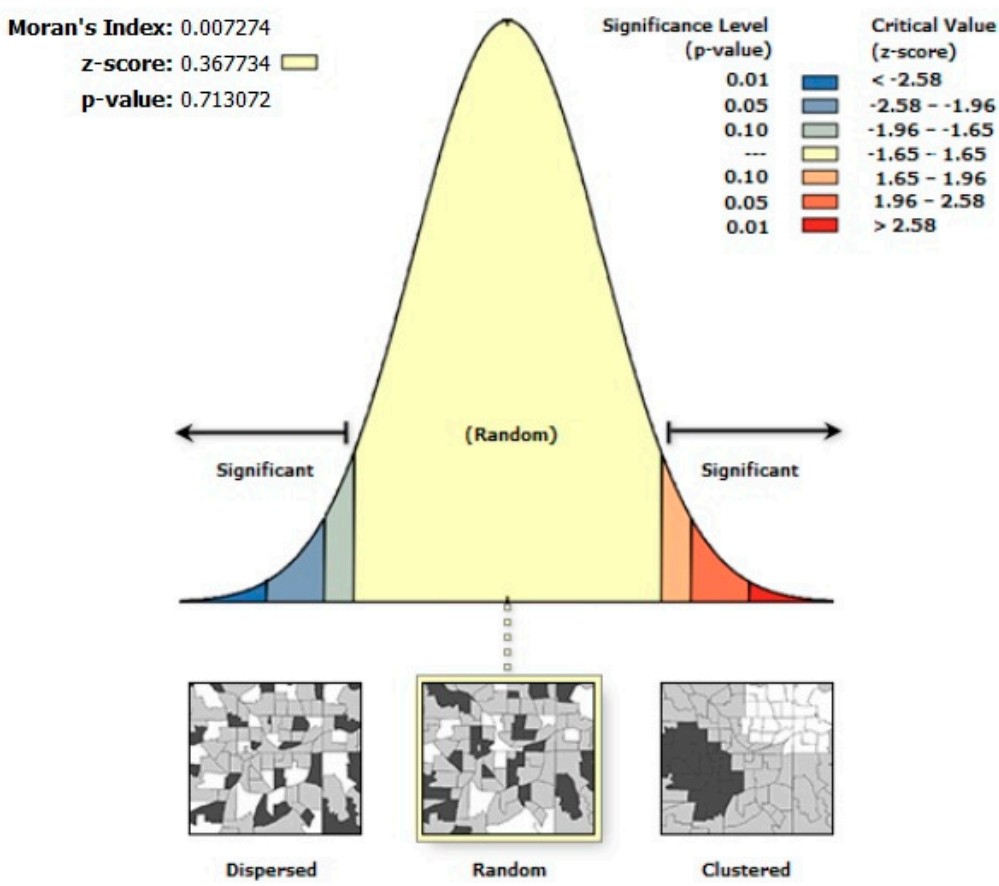

**Figure 12.** Non-Clustered Residual.

3.3.6. The Significance Level of Five Indicators Coefficient

Traffic Air Pollution Indicator (TAPI) was clustered more significantly toward alteration of SUTPI value in the north-east of Jakarta. It indicates that north-east area of Jakarta had high air pollution because of daily activity of commuter as gate to Jakarta in this area from Bekasi residency. The strategy of reducing air pollution was prioritized in the north-east region of Jakarta because according to the results of GWR analysis, it had a high coefficient compared to other regions in the Jakarta area as shown in Figure 14a. Traffic Noise Pollution Index (TNPI) was clustered highly in north of Jakarta. It means that it is more significant toward the alteration of SUTPI value in north of Jakarta. This indicates that the traffic noise pollution was caused by the activity of Tanjung Priok International port, offices, industrial zone, and business area. It makes the reference that the strategic policy in reducing noise pollution is by paying attention to the area with a high coefficient in the North of Jakarta (see Figure 14b).

**Table 9.** GRW Result of SUTPI in Jakarta city.

| FID | LocalR2 | Intercept | C1_Cong | C2_Accid | C3_AirPol | C4_Noise | C5_LandC | Residual |
|---|---|---|---|---|---|---|---|---|
| 1 | 0.872702 | −1.81482 | 0.398293 | 0.625328 | 0.390662 | 0.124234 | 0.094241 | 0.118103 |
| 2 | 0.872699 | −1.814932 | 0.398285 | 0.625399 | 0.390611 | 0.124261 | 0.094231 | 0.041828 |
| 3 | 0.872693 | −1.815171 | 0.398272 | 0.625521 | 0.390533 | 0.124324 | 0.094202 | −0.092107 |
| 4 | 0.872701 | −1.814736 | 0.398258 | 0.625529 | 0.390439 | 0.124171 | 0.094312 | 0.267413 |
| 5 | 0.872700 | −1.814975 | 0.398303 | 0.625296 | 0.390725 | 0.124294 | 0.094194 | −0.07683 |
| 6 | 0.872707 | −1.814515 | 0.398267 | 0.625431 | 0.390496 | 0.12411 | 0.094342 | −0.612814 |
| 7 | 0.872707 | −1.814522 | 0.398262 | 0.625461 | 0.390467 | 0.124108 | 0.094348 | −0.105381 |
| 8 | 0.872703 | −1.814734 | 0.398284 | 0.625369 | 0.390602 | 0.124197 | 0.094272 | −0.382117 |
| 9 | 0.872697 | −1.815018 | 0.39828 | 0.625446 | 0.390579 | 0.124283 | 0.094221 | 0.821003 |
| 10 | 0.872695 | −1.815078 | 0.398277 | 0.625475 | 0.390561 | 0.124299 | 0.094214 | 0.133799 |
| 11 | 0.872704 | −1.814596 | 0.398255 | 0.625517 | 0.390424 | 0.124124 | 0.094343 | 0.184042 |
| 12 | 0.872702 | −1.814801 | 0.398279 | 0.625412 | 0.390572 | 0.124213 | 0.094266 | 0.409616 |
| 13 | 0.872699 | −1.814946 | 0.398283 | 0.625414 | 0.390599 | 0.124264 | 0.094231 | 0.230304 |
| 14 | 0.872698 | −1.814997 | 0.398286 | 0.625404 | 0.390619 | 0.124284 | 0.094215 | −1.834353 |
| 15 | 0.872695 | −1.815044 | 0.398266 | 0.625537 | 0.390491 | 0.124277 | 0.094238 | −0.068336 |
| 16 | 0.872691 | −1.815257 | 0.398271 | 0.625547 | 0.390522 | 0.124349 | 0.094188 | −0.348562 |
| 17 | 0.872700 | −1.81485 | 0.398266 | 0.625498 | 0.390493 | 0.124216 | 0.094276 | 0.352316 |
| 18 | 0.872699 | −1.814864 | 0.398263 | 0.625522 | 0.390471 | 0.124217 | 0.094278 | 0.470292 |
| 19 | 0.872696 | −1.815033 | 0.398271 | 0.625505 | 0.390522 | 0.124279 | 0.094232 | 0.040886 |
| 20 | 0.872700 | −1.814939 | 0.398293 | 0.625353 | 0.39066 | 0.124272 | 0.094217 | 0.593639 |
| 21 | 0.872698 | −1.815029 | 0.398299 | 0.625333 | 0.390697 | 0.124307 | 0.09419 | 0.135157 |
| 22 | 0.872703 | −1.814689 | 0.39827 | 0.625447 | 0.390514 | 0.124168 | 0.094303 | 0.104081 |
| 23 | 0.872704 | −1.814684 | 0.398275 | 0.625416 | 0.390545 | 0.124172 | 0.094296 | 0.77437 |
| 24 | 0.872701 | −1.814807 | 0.398268 | 0.625481 | 0.390502 | 0.124204 | 0.094282 | −1.075187 |
| 25 | 0.872697 | −1.814994 | 0.398274 | 0.625479 | 0.390541 | 0.124269 | 0.094235 | 0.714113 |
| 26 | 0.872700 | −1.814878 | 0.398281 | 0.625414 | 0.390585 | 0.12424 | 0.094248 | 0.613541 |
| 27 | 0.872699 | −1.814934 | 0.398278 | 0.62544 | 0.390569 | 0.124255 | 0.09424 | 0.112377 |
| 28 | 0.872696 | −1.815102 | 0.398292 | 0.625388 | 0.390656 | 0.124323 | 0.094185 | −0.577856 |
| 29 | 0.872702 | −1.81477 | 0.39827 | 0.62546 | 0.390517 | 0.124195 | 0.094286 | 0.127757 |
| 30 | 0.872703 | −1.814697 | 0.398263 | 0.625491 | 0.390471 | 0.124164 | 0.094312 | −0.105086 |
| 31 | 0.872705 | −1.814567 | 0.398261 | 0.625479 | 0.390457 | 0.12412 | 0.094341 | 0.012695 |
| 32 | 0.872693 | −1.815219 | 0.398281 | 0.625475 | 0.390589 | 0.124349 | 0.094179 | 0.18828 |
| 33 | 0.872698 | −1.814889 | 0.39826 | 0.625545 | 0.390453 | 0.124222 | 0.094278 | −0.136275 |
| 34 | 0.872701 | −1.814874 | 0.398286 | 0.625384 | 0.390616 | 0.124244 | 0.094241 | −0.132374 |
| 35 | 0.872697 | −1.815044 | 0.398285 | 0.625423 | 0.390609 | 0.124297 | 0.094209 | −0.529793 |
| 36 | 0.872699 | −1.814951 | 0.398282 | 0.625424 | 0.390589 | 0.124264 | 0.094232 | −0.176698 |
| 37 | 0.872699 | −1.81488 | 0.398274 | 0.625457 | 0.390541 | 0.124233 | 0.094258 | −0.1323 |
| 38 | 0.872696 | −1.815078 | 0.398283 | 0.625435 | 0.390603 | 0.124306 | 0.094203 | −0.810122 |
| 39 | 0.872695 | −1.815098 | 0.398282 | 0.62545 | 0.390591 | 0.124311 | 0.094202 | 0.384754 |
| 40 | 0.872698 | −1.814941 | 0.398273 | 0.625472 | 0.390538 | 0.124252 | 0.094246 | −0.132303 |
| 41 | 0.872697 | −1.81504 | 0.398293 | 0.625371 | 0.390661 | 0.124304 | 0.094197 | 0.291371 |
| 42 | 0.872701 | −1.81483 | 0.398275 | 0.62544 | 0.390549 | 0.124219 | 0.094266 | 0.20609 |

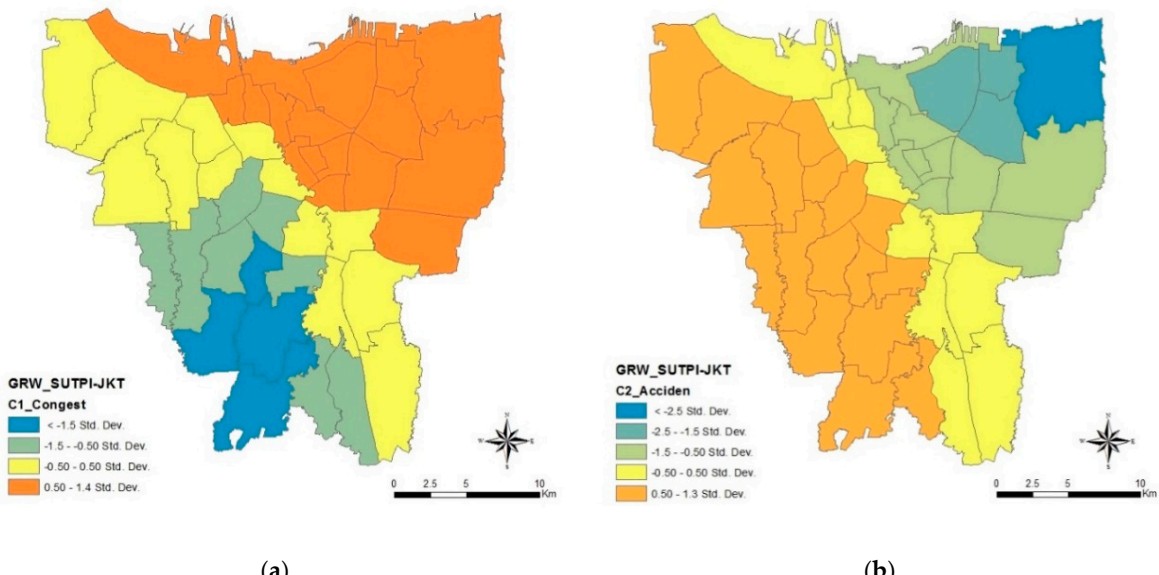

(**a**)          (**b**)

**Figure 13.** (**a**) Coefficient Significance of TCI (**b**) Coefficient Significance of TAccI.

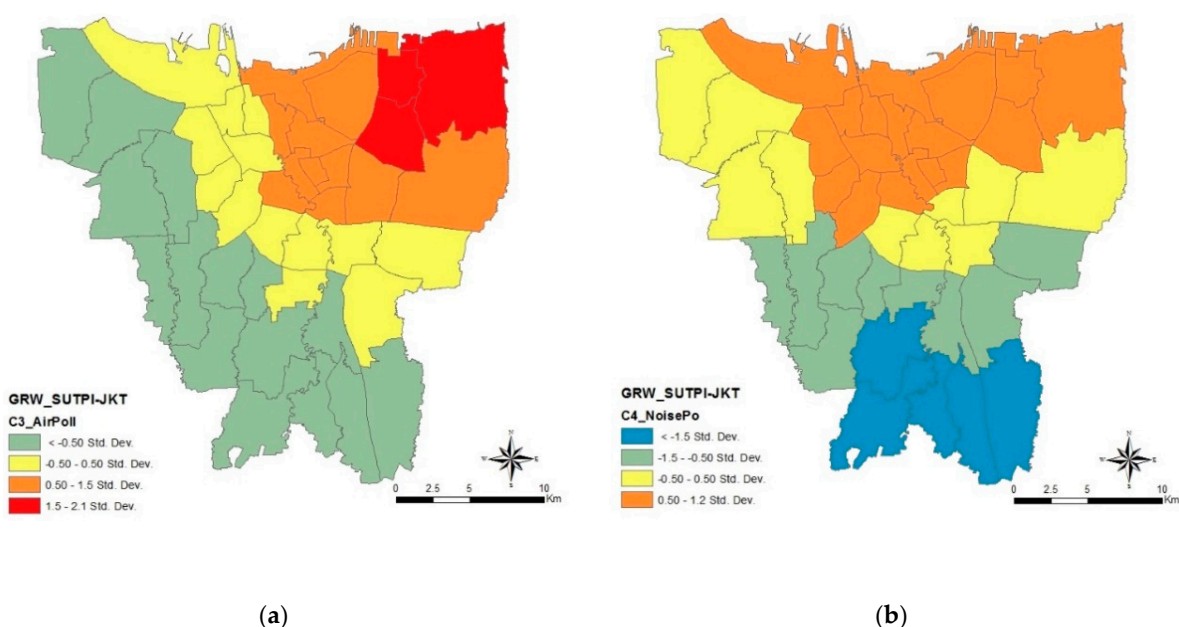

<p style="text-align:center;">(<b>a</b>)　　　　　　　　　　　　　　　　　　　　(<b>b</b>)</p>

**Figure 14.** (**a**) Coefficient Significance of TAPI (**b**) Coefficient Significance of TNPI.

Figure 15 describes that Land Consumption for Transport Infrastructure Index (TILCI) clustered is high in South Jakarta. It means that it is more significant toward alteration of SUTPI value in South Jakarta. High coefficient values for indicators of land consumption for SUT performance values were in the South area of Jakarta. While the highest usage indicator was in the central Jakarta. This means that in the future the performance of SUT in Jakarta will be significantly affected by land use in this area. Transport infrastructure was mostly higher in land in South area of Jakarta. Future planning strategies for land use in order to improve SUT performance is to pay attention in the South region of Jakarta, because this area has the most influence in increasing the value of SUT performance in the future.

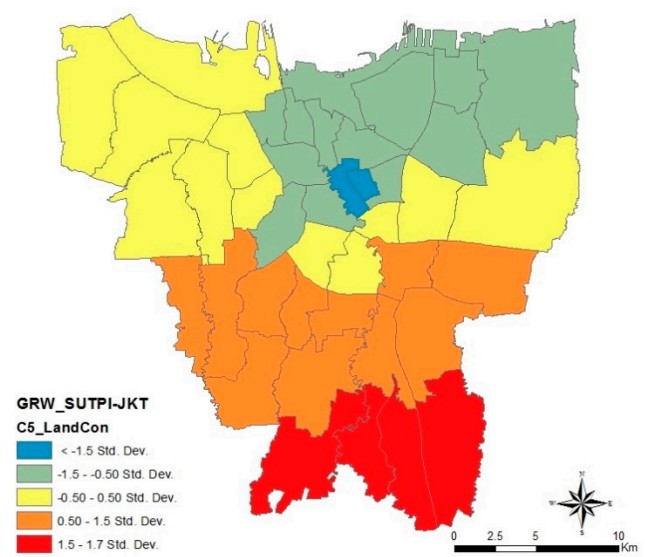

**Figure 15.** Coefficient Significance of TILCI.

### 3.3.7. The Predicted Performance of SUT in Jakarta city Based on GRW Analysis

Figure 16a depicts the result of composite SUT performance index which combines all the considered five indicators. It shows both dispersed and clustered patterns of performance of sustainable urban transport in Jakarta city. It is found that districts with high performance of SUT

were dispersed patterns in the city boundaries. Medium and low performance were concentrated and clustered in the North and East city, while some districts were found at the center of the city. It was significant that medium-low performance districts were dominant and spatially separated. Notably, about 13.76% of the Jakarta area were in the high performance level of SUT, about 22.22% were in the medium-high performance level, about 15.24% of Jakarta area were in the medium performance level, about 30.40% of Jakarta area were in the medium-low performance level, and about 18.38% of Jakarta area were in the low performance level.

Figure 16b describes the results of the GRW analysis in the form of prediction of the value of SUT performance. In general, the results of this spatial analysis show a decrease in the level of performance from the existing conditions of SUT performance. There were only a few districts in the East and South that still show high performance. The Comparison result between existing SUTPI and SUTPI prediction indicates a change in the composition of SUT district performance in Jakarta (see Table 10). For high performance from 13.76% area of Jakarta, it dropped to 11.04%. Medium-high performance from the existing conditions of 22.22% of the area decreased to 19.70% of the area. Whereas, medium performance from existing conditions of 15.24% of the area increased performance to 26.07% of the area. For medium-low performance of 30.40% of the area, it decreased to 25.96% of the area. Low performance decreased from 18.38% to 17.22% area. Based on the results of the GRW analysis by comparing it with the previous existing conditions, it shows a negative trend of SUT performance in Jakarta. Therefore, a strategic plan needs to be done by increasing efforts to overcome the problems of urban transportation which is in accordant with the results of the mapping of each indicator. This planning strategy takes into account the significance of indicators in each district, so that an increase in SUT performance becomes effective because it is in accordant with the identification of the main problems and their effects on improving SUT performance in each district and Jakarta as a whole.

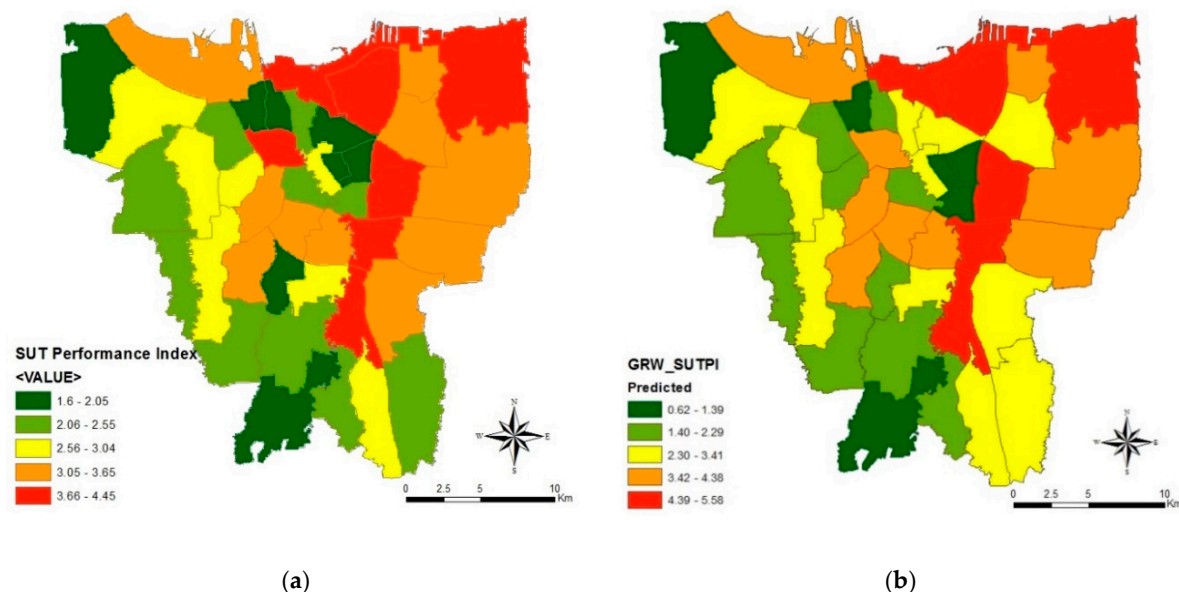

(**a**)          (**b**)

**Figure 16.** (**a**) SUTPI Existing (**b**) SUTPI Predicted.

### 3.3.8. GRW Model Validation of SUT in Jakarta city

The essential step in modelling methodology that the simulation data and the real data should be compared in order to validate the model and calculate the relative error and mean square deviation of the indicators [83]. In the model validation stage, there are two important aspects, namely the suitability of the behavior pattern between real data and the simulation results and about the close relationship between the values of real data and the simulation results. The pattern of suitability behavior reflects the model structure ability to mimic the behavior of the phenomenon that occurs. This model formation is the consequence of the displaying procedure of the causal impact relationship

of different affected segments. Whereas, the precision level and accuracy of the model are reflected by the closeness between the real data values and simulations results [84]. The examined variables on this study consist of five indicators: traffic congestion, traffic accident, traffic air pollution, traffic noise pollution, and land consumption for transport infrastructure. GRW analysis resulted the predictive model of SUTPI, three basic indicators were significant predictors, namely traffic congestion, traffic accident and traffic air pollution. Two basic indicators were not significant predictors, namely traffic noise pollution and land consumption. There was no multi-collinearity problem of the predictive model since the Variance Inflation Factor (VIF) ranged under 7.5. Furthermore, the adjusted R-square of the model was 0.845 indicating that 84.5% variability in SUTPI value could be explained by these variables.

**Table 10.** SUTPI Predicted Performance in Jakarta city based on GRW analysis.

| Performance | SUTPI | | District Name |
|---|---|---|---|
| | Total Dist. | Area (sqkm) | |
| High | 6 | 72.45 | Cempaka Putih, Jagakarsa, Johar Baru, Kalideres, Matraman, Tambora |
| Medium–High | 10 | 129.24 | Cilandak, Grogol Petamburan, Kembangan, Mampang Prapatan, Palmerah, Menteng, Pasar Minngu, Pasar Rebo, Pesanggrahan, Taman Sari |
| Medium | 11 | 171.06 | Cengkareng, Cipayung, Ciracas, Kemayoran, Kebayoran Lama, Kebon Jeruk, Kelapa Gading, Makasar, Pancoran, Sawah Besar, Senen |
| Medium–Low | 9 | 170.3 | Cakung, Duren Sawit, Gambir, Kebayoran Baru, Koja, Penjaringan, Setia Budi, Tanah Abang. Tebet |
| Low | 6 | 113 | Cilincing, Jatinegara, Kramat Jati, Pademangan, Pulo Gadung, Tanjung Priok |

Figure 17 represents comparison between actual SUTPI value and predicted SUTPI value. It can be interpreted that no significant differences in the SUTPI value in validation process because the predicted values of variables have the low relative errors of ±5% compared to the observed values based on Guan et al. [86]. This indicates the validity of model. So, the developed SUT model is reliable to elucidate the relationships and predict the dynamic change of JCR's SUT performance.

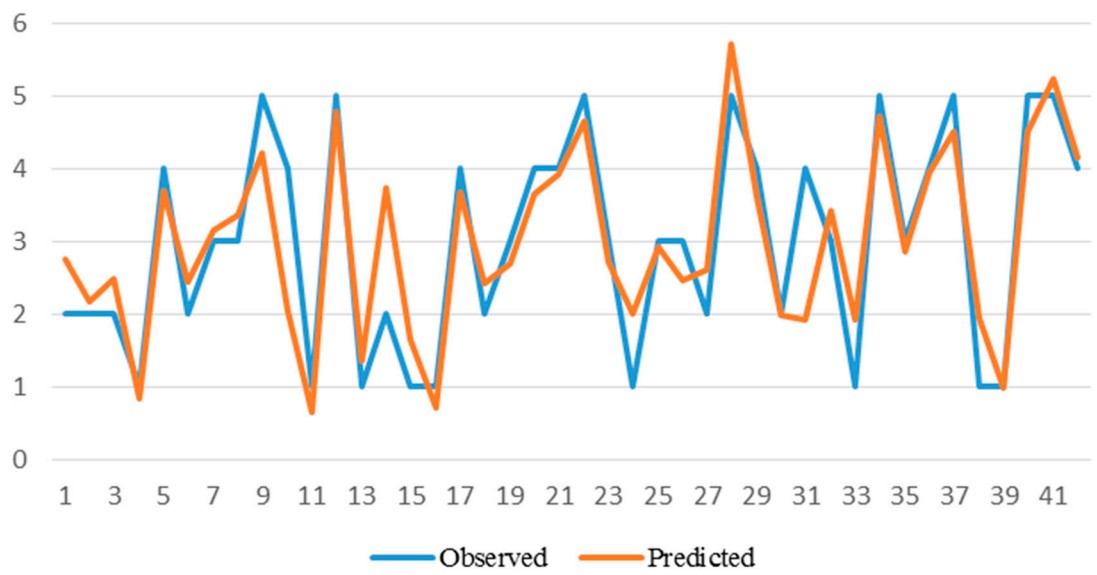

**Figure 17.** GRW Model Validation between SUTPI observed and SUTPI predicted.

The studies that are pure in exploring the measurement of SUT performance are still rare in recent times, especially those which use a quantitative approach and spatial analysis. For instance, there are current studies about sustainable urban transport index which have been developed by

UNESCAP [88] that introduce the measurements of SUT in four Asian cities, but the study results are still generally qualitative in describing the indicators presented by the spider diagram. In addition, the study explored by Doust and Parolin [89] using two indicators, namely accessibility and greenhouse gas emission in Sydney city, used the metric methodologies as the core the study for displaying the analysis result. Whereas, this study uses a simple approach, more specific and spatially, to assist the transport- and urban planners in identifying transportation issues for making strategic plans of SUT. Further studies can create SUT performance scenarios by first conducting the calibration process and clearly defining the sustainable urban transport performance index (SUTPI) by including measurement units to clarify how to read and interpret the results.

## 4. Conclusions

Sustainable Urban Transport Performance is how well urban Transport serves societies activities as a navigation tool to achieve goals of sustainability. This paper aims to model five basic indicators of SUT performance using GIS in Jakarta city. Urban transportation Jakarta should be controlled to keep it sustainable by managing it to remain effective and efficient. Therefore, performance measurements need to be done to find out the extent of current conditions, evaluate and control them for the future. In this study, the measurement of SUT performance of Jakarta city focuses on basic indicators such as traffic congestion, traffic accident, traffic pollution, traffic noise pollution, and land consumption for transport infrastructure. For a complete evaluation of system efficiency, all basic indicators have also to be considered together, not individually.

The measurement of sustainable urban transport performance is difficult through the evaluation of its indicators, but the Sustainable Urban Transport Performance Model (SUTPM) technique is able to prove to be a reliable method. This method is better than previous techniques which mostly counts one indicator of SUT and mostly qualitatively. The SUTPM method is designed to have more capability in measuring the performance of SUTs spatially and simply. The indicators used are by using basic indicator in local and regional scope. This model is to visualize the effect of the indicator on the performance of the SUT and its influence respectively. Therefore, this model can be used to measure and evaluate the development of urban transport in order to be sustainable.

Spatial patterns of travel demand and service distribution in each district are different. Each district has a performance value from the SUT to be measurable for its development. For the global study result indicates that the performance of SUT in Jakarta in medium level. The level of traffic congestion, air pollution and noise pollution continue to show increasing trends, caused by population growth, the rapid growth of the use of private vehicles and commuter from cities around Jakarta. Although, on the other side, accident rate indicates trends that continue to decrease and land use for road infrastructure in general is not excessive.

This research has investigated the sustainable urban transport conditions in the city of Jakarta with five basic indicators. It has identified the degree of sustainability in the urban transport system. Comprehensive and robust results can be obtained by analyzing five indicators: traffic congestion, accident rate, air pollution, noise pollution, and land use for transportation infrastructure. However, it is necessary to repair the completeness of the data and the latest version. The limitations in this research are: determining the efficiency level of SUT that is still under control with policy constrains, and the availability of comprehensive and up-to-date data using open data in the city of Jakarta is still limited. Some recommendations for further research development are making scenario evaluation, defining measurement unit of SUTPI, exploring big data and streaming data in transport website and media social, variation in using GIS tools, and using network analysis in modelling approach to get more specific in result and precise prediction.

**Author Contributions:** Conceptualization, P.A.N. and A.M.; methodology, P.A.N. and A.M.; software, P.A.N.; formal analysis, P.A.N.; investigation, P.A.N.; data curation, P.A.N.; writing—original draft preparation, P.A.N.; writing—review and editing, P.A.N. and A.M.; visualization, P.A.N.; supervision, A.M.

**Funding:** This work was funded by the Deanship of Scientific Research (DSR), King Abdulaziz University, Jeddah, Saudi Arabia.

**Acknowledgments:** This project was funded by the Deanship of Scientific Research (DSR), King Abdulaziz University, Jeddah, under grant No. (DG-030-137-1440). The authors, therefore, gratefully acknowledge the DSR technical and financial support.

**Conflicts of Interest:** The authors declare no conflict of interest.

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
