# Peer review of "Modelling Sustainable Urban Transport Performance in the Jakarta city Region: A GIS Approach"

_sustainability, doi:10.3390/su11071879_

Round 1
Reviewer 1 Report
This research article is very interesting. However, before accepting its publication it is necessary to make some changes:
1.- The cartography should be described in more detail (sources, precisions, etc).
2.- The selection of the variables should be justified
3.- The variables should be described in more detail. For example: Why did you measure congestion with TCI = Total accident / area (per district)? There are other more rigorous indicators to determine traffic congestion.
4.- It is necessary to explain the adjustment of the sensitivity of the weights in the regression model.
5.- Some statistical concepts should be reviewed. For example: the p-value proves that the model is significant, but not the intensity of the relationship between variables.
6.- In the Descriptive Statistical Analysis you only show the value of the statistical parameters, but not the real meaning.
7.- It would be convenient to compare the results obtained in the model with other regions to better understand the measurement of the Sustainable Urban Transport Performance in Jakarta.
8.- There are some problems with abbreviations, there are errors and this makes difficult the reading.
Author Response
Thank you for your comments.The answers has been attached.

Reviewer 2 Report
The paper reports an application for evaluating the sustainable urban transport by means of Geographic Information System (GIS).
A. State of the art
The state of the art reported in section 1 recalls many papers. The majority of them are relative to methods applied inside GIS (some other references are Borruso 2008, Transactions in GIS; Mohaymany et al. 2013, Geo-spatial Information Science; Okabe et al. Geographical Information Science). The main focus of the paper is on “Modelling sustainability urban transport performances ….”. They have to be evaluated with transport models. GIS is just a tool. For these reasons, the state of the art and the application should consider quantitative models for the simulation, forecasting and planning of transportation system (i.e. Ortuzar 2011, John Wiley & Sons; Cascetta 2013, Springer).
B. Data and model
Data are used for calculating sustainability indicators (i.e. Gillis et al. 2016, Sustainability). Their use is correct for historical and current evaluation. For the scenario evaluation, models calibrated in the present configuration are required (i.e. the difference between the use of big data and models is reported in Birgillito et al. 2018, JAT). Transport models are relative to: supply performances evaluation; demand and users’ behaviour analysis (only the level of trip generation is reported in line 118 adopting regression models instead of using random utility models); demand-supply interaction; system forecasting. After having applied these models, the forecasted indicators can be evaluated. Some references about transport models are reported in the above section A. The models adopted in the paper do not consider the models commented in the points A. and B..
C. Sustainability
Sustainability has three main pillars: environment, economic and social. The indicators adopted in the paper are five: congestion, accident, air, noise, land consumption. Many of the adopted indicators are relative to environment. For the analysis of sustainability, at least one indicator for each pillar can/should be considered. Each indicator needs to refer clearly to each pillar and has to describe a characteristic of the transport system (see, among the others, Sdoukopoulos et al. 2019, Transportation Research D; Croce et al., 2018, Advances in Modelling and Analysis A; Huovila et al. 2019, Cities)
D. Indicators
In the paper (Table 2) the total congestion is evaluated with total accident variables; the total accident is evaluated with total accident number. A correct measure of congestion is the ratio between traffic flow and traffic capacity on each link. Without using link traffic flow the congestion cannot be evaluated, with particular reference to the scenario configuration.
E. Accessibility
The accessibility in transport field is a measure of the capacity of traveller to reach a destination departing from an origin. In the accessibility indicator (also in relation to the active or passive specification of accessibility), socio-economic data, congested travel time, available destinations, …. can be considered (the indicators adopted in line 235-236 do not consider all these indicators and they are with a fixed weight).
F. SUTPI
For the calibration of eq. (1), SUTPI is the dependent variable. In the paper it is not reported as the SUTI is observed for calibrating the model. It cannot be considered as the combination of the independent variables. SUTPI is the main variable considered for measuring sustainability and, for the calibration issue, it has to be directly observed. SUTI has to be defined, indicating also the unit of measurement, in order to clarify how to read and interpret the results. Line 410 “almost 3, meaning in the middle performance level”: is arbitrary the definition of the level?
G. Prediction
Fig. 16(b) reports the predicted values. It is not reported how the scenario configuration is defined and the models adopted for evaluating the input variables in the scenario condition (the current observed variables cannot be adopted).
H. Minor
The order reported for indicator in the text line 61 and 64..73 are different.
The names of the five indicators are recalled in many points in the paper.
In line 154: sqkm. What does sq means?
In line 452 (fig. 10), considering the dispersion of the points, the calibration seem not linear in the specification.
Fig. 3 is in fig. 7.
Values of parameters are repeated in tab. 6 and eq. (3).
Author Response

(The authors gave the same response as above.)

Reviewer 3 Report
Dear Authors,
overall contribution is acceptable, and it ties with Sustainability.
However, some changes are necessary before I can recommend the paper for publication.

Author Response

(The authors gave the same response as above.)

Reviewer 4 Report
Interesting research, I like literature review, also very comprehensive database used. Minor negative is complexity of the paper and model/results presentation which makes it a bit difficult for readers. Some specific comments:
Why did you measure congestion with TCI = Total Accident / area (per district)? There are different interpretations of what congestion is, did you consider other factors (delay, density = Level Of Service)? EU project FLOW defines it like:
"Congestion is a state of traffic involving all modes on a multimodal transport
network (e.g. road, cycle facilities, pavements, bus lane) characterised by high densities and overused infrastructure compared to an acceptable state across all modes against previously-agreed targets and thereby leads to (perceived or actual) delay."
This approach requires quality indicators regarding performance, economics etc., safety is just one of the issues.
Did you test sensitivity of the model to other weights? (you have used traffic congestion by 25%, traffic accident by 30%, traffic air pollution by 25%, traffic noise pollution by 10% and land consumption by 10%)?
In literature, 42 elements are not considered valid to test regression with 6 independent variables (Tabachnick and Fidel (2013, p 123, give formula 50+8*number of variables = 102 for 6 variables). or you had more than 42 elements in your set? Scatter Plots Analysis shows not so strong relationship between variables, Please report R2 for each variable, you have reported only overall R2 to be more than 0.85.
It is stated that noise pollution TNPI is in negative relationship with SUTPI, which is true with scatter plot. Is there a mistake in sentence in line 446, which states that " Its means the coefficient represents that SUTPI increases as the traffic noise pollution increases ".
It is very interesting that there is no multi-collinearity between TCI = Total Accident / area (per district) and TaccI = Total Accident Number / area (per district), since both are based on accidents? (if I am not getting anything wrong). Please check this again or explain these variables more in detail.
Two indicators have no significant influence to SUTPI, could it be because of lower weight values in composite analysis? Did you try model excluding these two (backward regression)? Please see comment about sensitivity test.
I am curious how this model can be generalized. It is obvious that these regression coefficients and weights are typical for Jakarta, but looking forward model being tested in more urban areas.
Author Response

(The authors gave the same response as above.)

Round 2
Reviewer 1 Report
In my opinion, the article has been corrected sufficiently. Although the description of the methodology could be improved
Author Response
Thank you for your comments.
The answers has been attached.

Reviewer 2 Report
The manuscript has been improved, however some of the requests have been partially considered. The specific points partially considered are reported again below.
A. State of the art
The state of the art reported in section 1 recalls many papers. The majority of them are relative to methods applied inside GIS (some other references are Borruso 2008, Transactions in GIS; Mohaymany et al. 2013, Geo-spatial Information Science; Okabe et al. Geographical Information Science). The main focus of the paper is on “Modelling sustainability urban transport performances ….”. They have to be evaluated with transport models. GIS is just a tool. For these reasons, the state of the art and the application should consider quantitative models for the simulation, forecasting and planning of transportation system (i.e. Ortuzar 2011, John Wiley & Sons; Cascetta 2013, Springer).
B. Data and model
Data are used for calculating sustainability indicators (i.e. Gillis et al. 2016, Sustainability). Their use is correct for historical and current evaluation. For the scenario evaluation, models calibrated in the present configuration are required (i.e. the difference between the use of big data and models is reported in Birgillito et al. 2018, JAT). …...
C. Sustainability
Sustainability has three main pillars: environment, economic and social. ….. Many of the adopted indicators are relative to environment. For the analysis of sustainability, at least one indicator for each pillar can/should be considered. Each indicator needs to refer clearly to each pillar and has to describe a characteristic of the transport system (see, among the others, Sdoukopoulos et al. 2019, Transportation Research D; Croce et al., 2018, Advances in Modelling and Analysis A; Huovila et al. 2019, Cities)
F. SUTPI
For the calibration of eq. (1), SUTPI is the dependent variable. …… It cannot be considered as the combination of the independent variables. ……. SUTI has to be defined, indicating also the unit of measurement, in order to clarify how to read and interpret the results. ….
Author Response
Thank you for your comments, Sir.
The answers has been attached.

Reviewer 4 Report
Thank you for accepting previous comments.
Author Response
Comments and Suggestions for Author.
Thank you for accepting previous comments.
Response:
You are welcome, Sir.
Round 3
Reviewer 2 Report
The points indicated in my previous review report are considered only in some aspects.
The models described in point A and C are not considered.
The points B and F are considered for further researches.
Author Response

(The authors gave the same response as above.)

Round 4
Reviewer 2 Report
The authors have considered the majority of my requests, but some other requests are not completely addressed (i.e. point B and F). A different point of view is possible.
Author Response

(The authors gave the same response as above.)
